# Focal-SAM: Focal Sharpness-Aware Minimization for Long-Tailed Classification

**Sicong Li** [1 2]  **Qianqian Xu** [3]  **Zhiyong Yang** [4]  **Zitai Wang** [3]
**Linchao Zhang** [5]  **Xiaochun Cao** [6]  **Qingming Huang** [4 7 3]

## Abstract

Real-world datasets often follow a long-tailed distribution, making generalization to tail classes difficult. Recent methods resorted to long-tail variants of Sharpness-Aware Minimization (SAM), such as ImbSAM and CC-SAM, to improve generalization by flattening the loss landscape. However, these attempts face a trade-off between computational efficiency and control over the loss landscape. On the one hand, ImbSAM is efficient but offers only coarse control as it excludes head classes from the SAM process. On the other hand, CC-SAM provides fine-grained control through class-dependent perturbations but at the cost of efficiency due to multiple backpropagations. Seeing this dilemma, we introduce Focal-SAM, which assigns different penalties to class-wise sharpness, achieving fine-grained control without extra backpropagations, thus maintaining efficiency. Furthermore, we theoretically analyze Focal-SAM's generalization ability and derive a sharper generalization bound. Extensive experiments on both traditional and foundation models validate the effectiveness of Focal-SAM.

## 1. Introduction

In the past decades, deep learning has achieved remarkable success in various fields, including image classification (He et al., 2016), medical image processing (Ronneberger et al., 2015), and object detection (Ren et al., 2015). However, this success often relies on carefully curated, balanced datasets. In real-world scenarios, data often exhibits a *long-tailed* distribution, where a few categories have abundant samples while most categories contain only a small number of examples. Long-tailed learning focuses on effectively training models on such imbalanced datasets (Zhang et al., 2023; 2024a). Numerous approaches have been proposed to address this challenge, including re-sampling (Buda et al., 2018), re-balancing (Cui et al., 2019; Ren et al., 2020; Wang et al., 2023), representation learning (Zhu et al., 2022; Cui et al., 2024), ensemble learning (Wang et al., 2021; Zhang et al., 2022), and fine-tuning foundation models (Dong et al., 2023; Shi et al., 2024).

Recently, Rangwani et al. (2022) visualized the loss landscape of different classes and observed that tail classes often suffer from saddle points. Since the loss landscape is closely related to the generalization of modern neural networks (Keskar et al., 2017; Jiang et al., 2020), they apply Sharpness-Aware Minimization (SAM) (Foret et al., 2021) to help tail classes escape from saddle points. Later, since the original SAM operates on all classes, ImbSAM (Zhou et al., 2023a) excludes the head classes to better focus on flattening the landscape of the tail classes. However, when combined with popular re-balancing methods (Cao et al., 2019; Kini et al., 2021; Menon et al., 2021), this coarse-grained approach often overemphasizes the tail classes, leading to poor head and overall performance. To achieve fine-grained control, CC-SAM (Zhou et al., 2023b) uses class-dependent perturbation. However, the per-class perturbation requires at least $C$ additional backpropagations, where $C$ denotes the number of classes, making it rather computationally expensive. This raises a natural question: *Can we design a method that achieves both fine-grained control and computational efficiency?*

Targeting this goal, we integrate the focal mechanism (Lin et al., 2017) with SAM, inducing a novel approach named Focal-SAM. Specifically, we introduce the focal sharpness term, which is defined as the weighted sum of class-wise sharpness, where the weights decrease in a focal-like manner from head to tail classes. On the one hand, Focal-SAM controls the flatness of different classes in a fine-grained way, better balancing the performance between head and tail classes than ImbSAM, as shown in Fig.1(a). On the other

---

[1]Institute of Information Engineering, CAS [2]School of Cyber Security, University of Chinese Academy of Sciences [3]Key Lab. of Intelligent Information Processing, Institute of Computing Tech., CAS [4]School of Computer Science and Tech., University of Chinese Academy of Sciences [5]Artificial Intelligence Institute of China Electronics Technology Group Corporation, [6]School of Cyber Science and Tech., Shenzhen Campus of Sun Yat-sen University [7]BDKM, University of Chinese Academy of Sciences. Correspondence to: Qianqian Xu <xuqianqian@ict.ac.cn>, Qingming Huang <qmhuang@ucas.ac.cn>.

*Proceedings of the 42nd International Conference on Machine Learning*, Vancouver, Canada. PMLR 267, 2025. Copyright 2025 by the author(s).

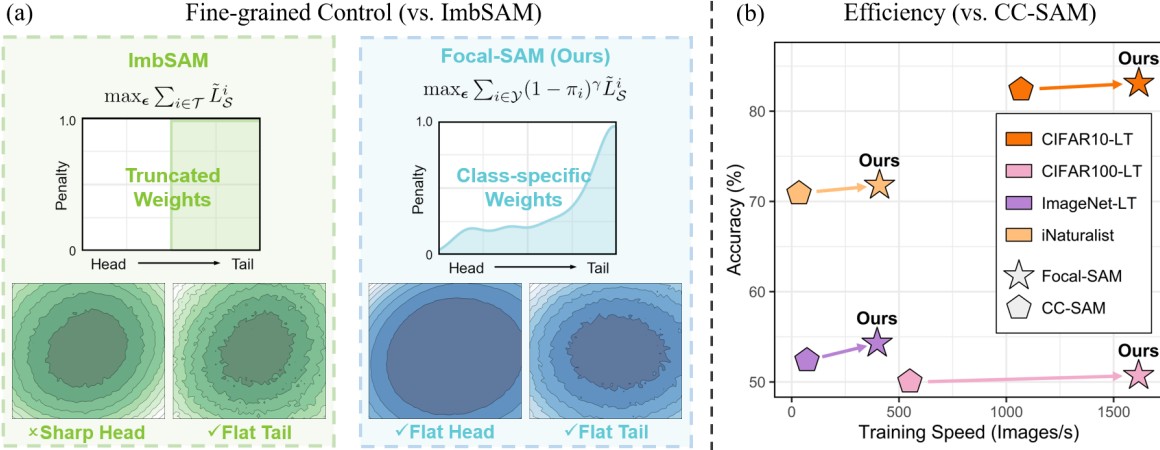

Figure 1: (a) ImbSAM applies the sharpness penalty only to tail classes, leading to a sharp loss landscape for head classes. In contrast, Focal-SAM assigns class-specific weights to the sharpness penalty, resulting in smooth loss landscapes for both head and tail classes. (b) Focal-SAM replaces per-class perturbations in CC-SAM with class-specific sharpness penalties, significantly enhancing computational efficiency while achieving better performance.

hand, Focal-SAM replaces the per-class perturbations in CC-SAM with per-class sharpness penalties, making it much more efficient than CC-SAM, as illustrated in Fig.1(b). Furthermore, we provide an informative generalization bound based on the PAC-Bayesian theory. This bound not only decreases at a faster rate than those of SAM and CC-SAM ($\tilde{\mathcal{O}}(1/n)$ vs. $\mathcal{O}(1/\sqrt{n})$, where $n$ is the number of training samples) but also demonstrates the influence of the hyperparameters and trace of the Hessian.

Finally, we conduct extensive experiments on various benchmark datasets to validate the effectiveness of Focal-SAM, including training ResNet models from scratch and finetuning the foundation model CLIP (Radford et al., 2021). The results show that Focal-SAM consistently outperforms other SAM-based methods across multiple datasets and models in long-tailed recognition tasks. Prior arts (Zhou et al., 2022; Khattak et al., 2023; Park et al., 2024) have demonstrated that fine-tuning CLIP often performs well on the target domain but struggles with domain shifts. Therefore, we also assess model performance on OOD test sets when fine-tuning foundation models, referred to as long-tailed domain generalization tasks. The results indicate that Focal-SAM improves performance by approximately 0.5%~4.3% when combined with baselines on OOD test sets. These further suggest that Focal-SAM can enhance generalization, leading to better performance under domain shifts.

In summary, our key contributions are as follows:

- Systematic studies illustrate the limitations of ImbSAM and CC-SAM. ImbSAM fails to flatten the loss landscape for head classes, while CC-SAM is highly computationally expensive.

- We propose Focal-SAM, a simple yet effective method that provides fine-grained control of loss landscape and maintains computational efficiency. Theoretical analysis further offers a sharp generalization bound of Focal-SAM.

- Extensive experiments validate the effectiveness of the proposed Focal-SAM, ranging from training ResNet models from scratch to fine-tuning foundation models.

## 2. Related Work

### 2.1. Long-Tailed Learning

Several approaches address long-tailed learning challenges, such as re-sampling (Buda et al., 2018; Wang et al., 2019b; Liu et al., 2022), re-balancing (Cui et al., 2019; Ren et al., 2020; Wang et al., 2023; 2022; Han et al., 2024; Hou et al., 2022; Lyu et al., 2025; Yang et al., 2023b;a; 2022; Zhao et al., 2024a; Dai et al., 2023; Shao et al., 2023; Hong et al., 2024), data augmentation (Kim et al., 2020; Hong et al., 2022; Ahn et al., 2023; Wang et al., 2024b;a), representation learning (Cui et al., 2021; Zhu et al., 2022; Cui et al., 2024; Gao et al., 2023; Zhang et al., 2024b), ensemble learning (Wang et al., 2021; Zhang et al., 2022; Li et al., 2022; Aimar et al., 2023; Yang et al., 2024; Zhao et al., 2024b), and fine-tuning foundation models (Dong et al., 2023; Shi et al., 2024). This paper focuses on loss modification, a technique that modifies the loss function to guide the model's attention towards tail classes, consequently improving their performance. Various methods have been proposed, such as LDAM (Cao et al., 2019), which enlarges the margin for tail classes to enhance their generalization performance. Cao et al. (2019) further introduce a training scheme called Deferred Re-weighting (DRW) used in conjunction with

LDAM to improve model performance. However, Menon et al. (2021) argue that previous loss modification techniques sacrifice consistency in minimizing the balanced error. They propose the LA (Menon et al., 2021) loss, which introduces adjustments to the standard cross-entropy loss to ensure Fisher consistency for balanced error minimization. Building on this work, the VS (Kini et al., 2021) loss further improves upon the LA loss by incorporating both additive and multiplicative adjustments, beneficial during the initial and terminal phases of training respectively. Most recently, Wang et al. (2023) provide a comprehensive generalization analysis of these losses.

In this paper, we leverage these loss functions while aiming to specifically improve their generalization ability for long-tailed classification tasks.

## 2.2. Sharpness of Loss Landscape

Generalization in deep neural networks has always been a crucial focus in machine learning research. Recent studies (Keskar et al., 2017; Jiang et al., 2020) have empirically and theoretically demonstrated that flatter minima in the loss landscape typically lead to better generalization. Inspired by this, Sharpness-Aware Minimization (SAM) (Foret et al., 2021) is developed to find flatter minima, achieving superior performance across various tasks.

In the context of long-tailed learning, Rangwani et al. (Rangwani et al., 2022) suggest combining SAM with re-balancing techniques to help the model escape saddle points and improve generalization. Imbalanced SAM (ImbSAM) (Zhou et al., 2023a) incorporates class priors into SAM by dividing classes into head and tail groups. It applies SAM exclusively to the tail classes while maintaining standard optimization for head classes, aiming to specifically enhance the generalization of tail classes. Class-Conditional SAM (CC-SAM) (Zhou et al., 2023b) applies SAM to each class individually, using class-specific perturbation radii. These radii increase from head to tail classes, enabling fine-grained control over the loss landscape for each class.

This work also extends the SAM framework for long-tailed classification. Our method aims to achieve fine-grained control over the loss landscape while maintaining computational efficiency.

## 3. Motivation

### 3.1. Problem Setup

We define the sample space as $\mathcal{X}$ and the label space as $\mathcal{Y} = \{1, 2, \cdots, C\}$. In the long-tailed recognition task, the training set follows an imbalanced distribution $\mathcal{D}$ and consists of data pairs denoted as $\mathcal{S} = \{(\boldsymbol{x}_i, y_i)\}_{i=1}^n$, where $y_i \in \mathcal{Y}$ is the label for sample $\boldsymbol{x}_i \in \mathcal{X}$, and $n$ is the to-

tal number of training samples. Let $\mathcal{D}_{bal}$ denote the uniform test distribution. Following prior work (Cao et al., 2019; Hong et al., 2021), given a class $y$, $\mathcal{D}$ and $\mathcal{D}_{bal}$ share the same class-conditional distribution, denoted as $\mathcal{D}_y \triangleq P(\boldsymbol{x}|y)$. We use $n_y$ to represent the number of samples in the $y$-th class and $\pi_y = n_y/n$ to denote the ratio of the $y$-th class in the training set. Without loss of generality, we assume $n_1 \geq n_2 \geq \cdots \geq n_C$, with $n_1 \gg n_C$.

The model parameters are denoted by $\boldsymbol{w}$, with a total of $k$ parameters. The loss for sample $(\boldsymbol{x}, y)$ is defined as $\ell(\boldsymbol{w}; \boldsymbol{x}, y)$. The training loss over dataset $\mathcal{S}$ is given by $L_S(\boldsymbol{w}) \triangleq \frac{1}{n} \sum_{i=1}^n \ell(\boldsymbol{w}; \boldsymbol{x}_i, y_i)$. Similarly, the loss specifically for samples from the $y$-th class within $\mathcal{S}$ is defined as $L_S^y(\boldsymbol{w}) \triangleq \frac{1}{n} \sum_{y_i=y} \ell(\boldsymbol{w}; \boldsymbol{x}_i, y_i)$. We further define the expected loss over $\mathcal{D}$, $\mathcal{D}_{bal}$ and $\mathcal{D}_y$ as $L_{\mathcal{D}}(\boldsymbol{w}) \triangleq \mathbb{E}_{(\boldsymbol{x},y) \sim \mathcal{D}}[\ell(\boldsymbol{w}; \boldsymbol{x}, y)]$, $L_{\mathcal{D}_{bal}}(\boldsymbol{w}) \triangleq \mathbb{E}_{(\boldsymbol{x},y) \sim \mathcal{D}_{bal}}[\ell(\boldsymbol{w}; \boldsymbol{x}, y)]$ and $L_{\mathcal{D}_y}(\boldsymbol{w}) \triangleq \mathbb{E}_{\boldsymbol{x} \sim \mathcal{D}_y}[\ell(\boldsymbol{w}; \boldsymbol{x}, y)]$, respectively. Our goal is to optimize parameters $\boldsymbol{w}$ on dataset $\mathcal{S}$ such that $L_{\mathcal{D}_{bal}}(\boldsymbol{w})$ is minimized, leading to good performance on the balanced test set.

### 3.2. Limitations in ImbSAM and CC-SAM

**ImbSAM.** ImbSAM divides classes into head and tail groups, denoted as $\mathcal{H}$ and $\mathcal{T}$. It applies SAM only to the tail group to focus on flattening loss landscape for these classes. Its objective function is:

$$L_S^{IS}(\boldsymbol{w}) \triangleq L_S^{\mathcal{H}}(\boldsymbol{w}) + \max_{\|\boldsymbol{\epsilon}\|_2 \leq \rho} L_S^{\mathcal{T}}(\boldsymbol{w} + \boldsymbol{\epsilon}) \quad (1)$$

From Eq.(1), ImbSAM excludes all head classes from SAM. As a result, **the loss landscape for head classes becomes sharper**, which may reduce their generalization performance. To validate this, we analyze the spectral density of the Hessian $H$ (Ghorbani et al., 2019), a common measure for the flatness of the loss landscape. We also consider two key metrics: the largest eigenvalue $\lambda_{max}$ and the trace $Tr(H)$. Higher values of $\lambda_{max}$ and $Tr(H)$ generally indicate a sharper loss landscape. Following prior work (Rangwani et al., 2022), we compute the eigen spectral density of the Hessian for head and tail classes on the CIFAR-10 LT dataset using the VS loss function. The results are shown in Fig.2.

A comparison between Fig.2(e) and Fig.2(f) reveals that ImbSAM effectively reduces $Tr(H)$ and $\lambda_{max}$ for the tail classes, suggesting a flatter loss landscape. However, when comparing Fig.2(a) and Fig.2(b), we observe that with ImbSAM, the values of $Tr(H)$ and $\lambda_{max}$ for head classes are significantly higher. This indicates that ImbSAM's exclusion of head classes from SAM sharpens their loss landscape, potentially degrading their generalization performance.

**CC-SAM.** CC-SAM applies SAM to each class individually, using class-specific perturbation radii. The objective

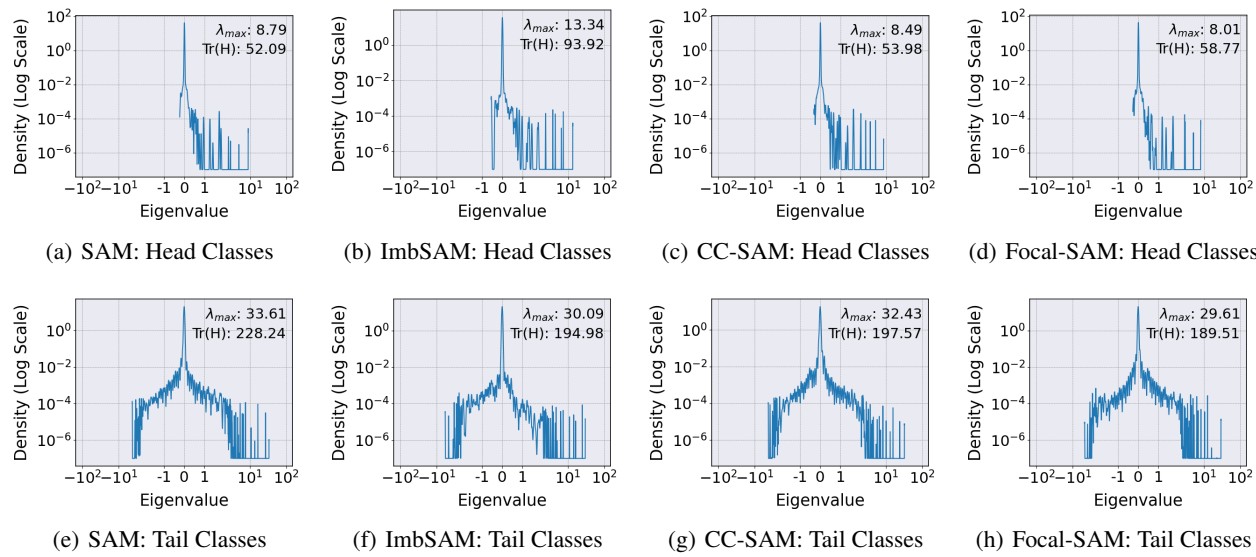

Figure 2: Eigen Spectral Density of Hessian for head and tail classes of ResNet models trained with various SAM variants on CIFAR-10 LT using VS loss. A smaller $\lambda_{max}$ and $Tr(H)$ generally indicate a flatter loss landscape.

Table 1: Average training time per epoch (in seconds) for different SAM variants across four long-tailed datasets using ResNet models. For CC-SAM, we follow its protocol by perturbing only the last few layers to improve its efficiency.

| Methods | CIFAR-10 LT | CIFAR-100 LT | ImageNet-LT | iNaturalist |
|---|---|---|---|---|
| SAM | 5.66s (1.00×) | 4.81s (1.00×) | 170.04s (1.00×) | 831.67s (1.00×) |
| ImbSAM | 7.80s (1.37×) | 6.68s (1.39×) | 293.11s (1.72×) | 1088.61s (1.31×) |
| CC-SAM | 11.61 (2.05×) | 19.70s (4.10×) | 1626.54s (9.57×) | 12869.89s (15.47×) |
| **Focal-SAM (Ours)** | 7.67s (1.36×) | 6.71s (1.40×) | 291.05s (1.71×) | 1068.92s (1.29×) |

function is defined as:

$$L_S^{CS}(\boldsymbol{w}) \triangleq \sum_{i=1}^{C} \max_{\|\boldsymbol{\epsilon}\|_2 \leq \rho_i^*} \frac{1}{\pi_i} \cdot L_S^i(\boldsymbol{w} + \boldsymbol{\epsilon}) \quad (2)$$

The optimal perturbation $\hat{\boldsymbol{\epsilon}}_i(\boldsymbol{w})$ for each class $i$ is also class-wise and estimated as $\rho_i^* \nabla_{\boldsymbol{w}} L_S^i(\boldsymbol{w}) / \|\nabla_{\boldsymbol{w}} L_S^i(\boldsymbol{w})\|_2$. The model parameters are updated with the learning rate $\eta$ as:

$$\boldsymbol{w} \leftarrow \boldsymbol{w} - \eta \sum_{i=1}^{C} \frac{1}{\pi_i} \cdot \nabla_{\boldsymbol{w}} L_S^i(\boldsymbol{w})|_{\boldsymbol{w} + \hat{\boldsymbol{\epsilon}}_i(\boldsymbol{w})} \quad (3)$$

This fine-grained method flattens the loss landscape of head and tail classes more effectively, as shown in Fig.2(c) and Fig.2(g). However, **CC-SAM is much more computationally demanding than SAM**. According to Eq.(3), per parameters update requires computing the gradient for each class $i$'s loss at $\boldsymbol{w} + \hat{\boldsymbol{\epsilon}}_i(\boldsymbol{w})$, i.e., $\nabla_{\boldsymbol{w}} L_S^i(\boldsymbol{w})|_{\boldsymbol{w} + \hat{\boldsymbol{\epsilon}}_i(\boldsymbol{w})}$. Therefore, CC-SAM requires at least $C$ backpropagations per update, whereas SAM only needs two. Thus, CC-SAM has a much higher computational cost than SAM. For details on the backpropagation requirements for SAM and ImbSAM, please see App.B.

To confirm this, we measure the average training time per epoch for various SAM variants across four datasets using ResNet models. For CC-SAM, we follow its protocol by perturbing only the last few layers to enhance efficiency. As shown in Tab.1, despite perturbing fewer parameters, CC-SAM takes about 2∼15× more time than SAM, depending on the dataset. The training time ratio of CC-SAM to SAM grows with the number of classes in the batch. These high computational costs make CC-SAM particularly impractical for large-scale datasets or fine-tuning foundation models.

## 4. Methodology

### 4.1. Focal Sharpness-Aware Minimization

Motivated by the analysis in Sec.3, we develop a new method called Focal-SAM. This approach achieves fine-grained control over the flatness between head and tail classes while maintaining computational efficiency, as shown in Tab.1 and Fig.2.

To this end, we first introduce the concept of **class-wise**

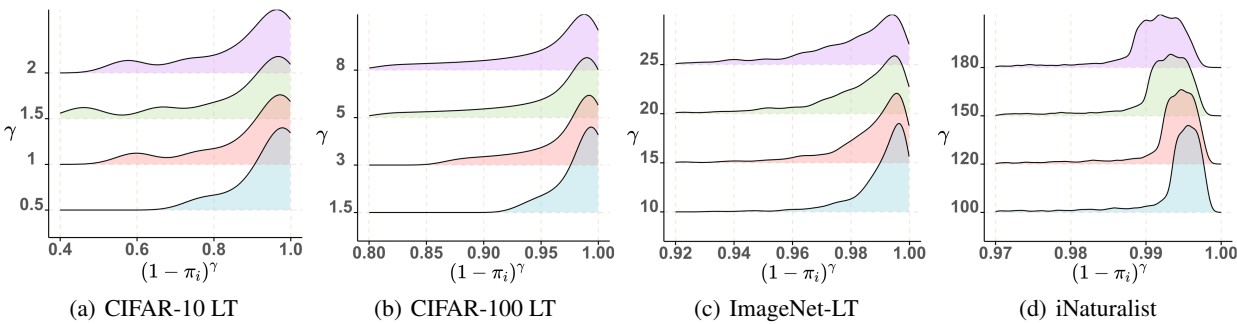

(a) CIFAR-10 LT  (b) CIFAR-100 LT  (c) ImageNet-LT  (d) iNaturalist

Figure 3: The probability density distributions of $(1 - \pi_i)^\gamma$ for various $\gamma$ values on CIFAR-10 LT, CIFAR-100 LT, ImageNet-LT, and iNaturalist.

**sharpness**, defined as the loss difference between the original model parameters $\boldsymbol{w}$ and the perturbed ones, to quantify the sharpness of loss landscapes across different classes:

$$\tilde{L}_S^i(\boldsymbol{w}, \boldsymbol{\epsilon}) \triangleq L_S^i(\boldsymbol{w} + \boldsymbol{\epsilon}) - L_S^i(\boldsymbol{w}), i \in \mathcal{Y}. \quad (4)$$

Next, we propose a new sharpness term called **focal sharpness**:

$$\tilde{L}_S^{FS}(\boldsymbol{w}) = \max_{\|\boldsymbol{\epsilon}\|_2 \leq \rho} \sum_{i=1}^{C} (1 - \pi_i)^\gamma \tilde{L}_S^i(\boldsymbol{w}, \boldsymbol{\epsilon}), \quad (5)$$

where $(1-\pi_i)^\gamma$ is the focal weight that provides fine-grained control over class-wise sharpness, and $\gamma$ is a tunable hyperparameter. When $\gamma$ increases, the distribution of focal weight $(1 - \pi_i)^\gamma$ will skew more to tail classes. Fig.3 illustrates how the probability density distributions of $(1 - \pi_i)^\gamma$ varies with respect to $\gamma$ on various long-tailed datasets.

Then, the objective of Focal-SAM is defined by the combination of the training loss and the focal sharpness term:

$$L_S^{FS}(\boldsymbol{w}) = L_S(\boldsymbol{w}) + \lambda \cdot \tilde{L}_S^{FS}(\boldsymbol{w}), \quad (6)$$

where $\lambda$ is a hyperparameter controlling the importance of focal sharpness. This formulation highlights how Focal-SAM overcomes ImbSAM's limitations. When $\gamma = 0$ and $\lambda = 1$, Eq.(5) penalizes the sharpness of each class equally, reverting to standard SAM. Conversely, when $\gamma$ is sufficiently large, focal weights for head classes rapidly approach 0, while the weights for tail classes remain relatively large. In this scenario, Focal-SAM approximates ImbSAM. Typically, we select a moderate $\gamma$, such that the focal weights increase smoothly from head to tail classes. This fine-grained control over loss landscape improves the flatness of tail classes while maintaining that of head classes, ultimately enhancing generalization for both traditional and foundation models.

### 4.2. Optimizing the Focal-SAM Objective Function

In this section, we discuss how to optimize the Focal-SAM objective $L_S^{FS}(\boldsymbol{w})$. Let $L_S^\gamma(\boldsymbol{w}) \triangleq \sum_{i=1}^{C}(1 - \pi_i)^\gamma L_S^i(\boldsymbol{w})$.

Using a first-order Taylor expansion, we approximate the solution of the inner maximization problem for $\tilde{L}_S^{FS}(\boldsymbol{w})$:

$$\hat{\boldsymbol{\epsilon}}(\boldsymbol{w}) \approx \underset{\|\boldsymbol{\epsilon}\|_2 \leq \rho}{\mathrm{argmax}} \, \boldsymbol{\epsilon}^T \nabla_{\boldsymbol{w}} L_S^\gamma(\boldsymbol{w}) = \rho \frac{\nabla_{\boldsymbol{w}} L_S^\gamma(\boldsymbol{w})}{\|\nabla_{\boldsymbol{w}} L_S^\gamma(\boldsymbol{w})\|_2} \quad (7)$$

Then, we can substitute $\boldsymbol{\epsilon}$ and compute the gradients of $L_S^{FS}(\boldsymbol{w})$ to solve the outer minimization problem:

$$\begin{aligned}
\nabla_{\boldsymbol{w}} L_S^{FS}(\boldsymbol{w}) &\approx \nabla_{\boldsymbol{w}}\big(L_S(\boldsymbol{w}) + \lambda[L_S^\gamma(\boldsymbol{w} + \hat{\boldsymbol{\epsilon}}(\boldsymbol{w})) - L_S^\gamma(\boldsymbol{w})]\big) \\
&\approx \nabla_{\boldsymbol{w}}\big(L_S(\boldsymbol{w}) - \lambda L_S^\gamma(\boldsymbol{w})\big)\big|_{\boldsymbol{w}} + \lambda \nabla_{\boldsymbol{w}} L_S^\gamma(\boldsymbol{w})\big|_{\boldsymbol{w}+\hat{\boldsymbol{\epsilon}}(\boldsymbol{w})}
\end{aligned}$$
$$(8)$$

From Eq.(7) and Eq.(8), computing $\nabla_{\boldsymbol{w}} L_S^{FS}(\boldsymbol{w})$ to update model parameters requires only three backpropagations: one for $\nabla_{\boldsymbol{w}} L_S^\gamma(\boldsymbol{w})$, one for $\nabla_{\boldsymbol{w}}(L_S(\boldsymbol{w}) - \lambda L_S^\gamma(\boldsymbol{w}))|_{\boldsymbol{w}}$, and one for $\nabla_{\boldsymbol{w}} L_S^\gamma(\boldsymbol{w})|_{\boldsymbol{w}+\hat{\boldsymbol{\epsilon}}(\boldsymbol{w})}$. Therefore, Focal-SAM is more computationally efficient than CC-SAM, making it more suitable for large-scale datasets or fine-tuning foundation models.

Overall, Alg.1 gives the pseudo-code to optimize the Focal-SAM objective, using SGD as the base optimizer.

---

**Algorithm 1** Focal-SAM algorithm

---

**Input:** Training set $S$, perturbation radius $\rho$, hyperparameter $\lambda$, $\gamma$, learning rate $\eta$
**Output:** Model trained with Focal-SAM
1: Initialize weights $\boldsymbol{w}_0$, $t = 0$;
2: **while** *not converged* **do**
3:    Sample batch $B = \{(\boldsymbol{x}_1, y_1), \cdots, (\boldsymbol{x}_b, y_b)\}$;
4:    Compute $L_B^\gamma(\boldsymbol{w})$;
5:    Compute $\nabla_{\boldsymbol{w}} L_B^\gamma(\boldsymbol{w})$ and $\hat{\boldsymbol{\epsilon}}(w)$ according to Eq.(7);
6:    Perturb $\boldsymbol{w}$ with $\hat{\boldsymbol{\epsilon}}(\boldsymbol{w})$, and compute gradient $\boldsymbol{g}_1 = \nabla_{\boldsymbol{w}} L_B^\gamma(\boldsymbol{w})|_{\boldsymbol{w}+\hat{\boldsymbol{\epsilon}}(\boldsymbol{w})}$;
7:    Compute gradient $\boldsymbol{g}_2 = \nabla_{\boldsymbol{w}}[L_B(\boldsymbol{w}) - \lambda \cdot L_B^\gamma(\boldsymbol{w})]|_{\boldsymbol{w}}$;
8:    Update weights: $\boldsymbol{w}_{t+1} = \boldsymbol{w}_t - \eta(\lambda \boldsymbol{g}_1 + \boldsymbol{g}_2)$;
9:    $t = t + 1$;
10: **end while**

---

### 4.3. Generalization Ability of Focal-SAM

Previous works have established the generalization bound for SAM (Foret et al., 2021) and CC-SAM (Zhou et al., 2023b). However, these bounds are relatively loose (with an order of $1/\sqrt{n}$) and could bias the training process. For example, the perturbation radius of CC-SAM (*i.e.*, $\rho_i$ in Eq.(2)) is set as the solution to minimizing its PAC-Bayesian bound. Since the generalization is not sharp enough, the estimated perturbation radius $\rho_i^*$ could deviate from the optimal one, thus leading to inferior performance. In this section, we develop a sharper generalization bound with an order of $1/n$ for Focal-SAM.

We assume the loss function $\ell(\boldsymbol{w}; \boldsymbol{x}, y)$ has an upper bound of $B$, which is a common and practical assumption. Then, we derive the following generalization bound based on the PAC-Bayesian theorem proposed in (Tolstikhin & Seldin, 2013). For conciseness, we present an informal formulation in the main content, leaving the formal one and the corresponding proof in App.A.

**Theorem 4.1** (**Informal**). *Assume that $\forall (\boldsymbol{x}, y) \in \mathcal{D}, 0 \leq \ell(\boldsymbol{w}; \boldsymbol{x}, y) \leq B$. For any $\rho > 0$, any uniform distribution $\mathcal{D}_{bal}$ and any distribution $\mathcal{D}$, with high probability over the choice of the training set $S \sim \mathcal{D}$,*

$$L_{\mathcal{D}_{bal}}(\boldsymbol{w}) \leq \underbrace{\frac{2L_S^{FS}(\boldsymbol{w})}{C\pi_C}}_{\text{(I)}} - \underbrace{\mathcal{O}\left(\frac{\lambda\rho^2}{k + \ln(n)} \cdot tr(H(\boldsymbol{w}))\right)}_{\text{(II)}}$$
$$+ \underbrace{\tilde{\mathcal{O}}\left(\frac{\lambda\left[k\log(\|\boldsymbol{w}\|_2^2/\rho^2) + \Psi\right]}{n}\right)}_{\text{(III)}}.$$
$$\tag{9}$$

*where $n = |S|$, $\Psi \triangleq \sum_{i=1}^{C}(1 - \pi_i)^\gamma \pi_i$, $k$ is the number of parameters, $H(\boldsymbol{w})$ represents the Hessian matrix of $L_{\mathcal{D}}^\gamma(\boldsymbol{w})$ at point $\boldsymbol{w}$ and $tr(\cdot)$ represents the matrix trace.*

From the theorem, we have the following insights:

- The generalization bound consists of three components. Specifically, (I) is the empirical loss on the training set $L_S^{FS}(\boldsymbol{w})$, which can be minimized via large-scale models. (II) reveals how the generalization performance is affected by multiple factors, including $\lambda, \rho, tr(H(\boldsymbol{w}))$. (III) decreases at a faster rate of $\tilde{\mathcal{O}}(1/n)$.

- The hyperparameters $\lambda$ and $\gamma$ play a crucial role. On the one hand, a larger $\lambda$ can increase both components (II) and (III) of the bound. Therefore, careful tuning of $\lambda$ can induce a tighter bound. On the other hand, a larger $\gamma$ leads to a smaller $\Psi$, also leading to a tighter bound. This suggests that assigning greater weights to the sharpness of the tail classes can improve the overall generalization ability.

- Focal-SAM enables a more effective optimization of $L_{\mathcal{D}_{bal}}(\boldsymbol{w})$. Specifically, we can reformulate Eq.(9) to

$$L_{\mathcal{D}_{bal}}(\boldsymbol{w}) + \text{(II)} \leq \text{(I)} + \text{(III)}. \tag{10}$$

Typically, (II) tends to be large without SAM-based techniques. As a result, minimizing the right-hand side (RHS) of Eq.(10) in such cases may not induce a small $L_{\mathcal{D}_{bal}}(\boldsymbol{w})$. In contrast, Focal-SAM reduces the trace $tr(H(\boldsymbol{w}))$ by effectively flattening the loss landscape, leading to a small (II). This makes it more effective to minimize $L_{\mathcal{D}_{bal}}(\boldsymbol{w})$ when we optimize the RHS of Eq.(10). This insight again validates the necessity of Focal-SAM.

## 5. Experiments

This section evaluates the effectiveness of Focal-SAM through a series of experiments. **Detailed experimental settings and additional results are provided in App.C and App.D due to space constraints.**

### 5.1. Experiment Protocols

**Datasets.** We use four widely adopted long-tailed datasets for long-tailed recognition tasks: CIFAR-10 LT (Cao et al., 2019), CIFAR-100 LT (Cao et al., 2019), ImageNet-LT (Liu et al., 2019) and iNaturalist (Horn et al., 2018). The CIFAR-LT datasets include variants with imbalance ratios of {200, 100, 50, 10}. In addition to evaluating model performance on ID test sets, we also assess it on **OOD** test sets, referred to as long-tailed domain generalization tasks. Specifically, we train the model on ImageNet-LT and evaluate it on three OOD datasets: ImageNet-Sketch (Wang et al., 2019a), ImageNetV2 (Recht et al., 2019), and ImageNet-C (Hendrycks & Dietterich, 2019). For more details, see App.C.1.

**Competitors.** When training ResNet models on the CIFAR-LT dataset, we assess several loss functions. These methods are further combined with SAM (Foret et al., 2021), ImbSAM (Zhou et al., 2023a), and CC-SAM (Zhou et al., 2023b) as baselines. For the ImageNet-LT and iNaturalist datasets, we employ a range of representative methods as baseline methods. When fine-tuning the foundation model CLIP (Radford et al., 2021), we evaluate both full fine-tuning with LA loss (denoted as FFT) and parameter-efficient fine-tuning using the LIFT method (Shi et al., 2024), along with their performance when combined with different SAM variants. For more details, please refer to App.C.2.

**Evaluation Protocol.** For long-tailed recognition tasks, we assess model performance using balanced accuracy (Menon et al., 2021). To provide deeper insights, we split the classes into three groups: Head, Medium, and Tail, and report accuracy for each group individually. For long-tailed domain generalization tasks, we evaluate performance on OOD balanced test sets, including top-1 accuracy and accuracy for

Table 2: Performance comparison on CIFAR-100 LT datasets with various imbalance ratios (IR). FFT denotes fully fine-tuning the foundation model with LA loss. **Due to space limitations, additional CIFAR-100 LT results combining more methods, as well as the CIFAR-10 LT results, are shown in Tab.6 and Tab.5.**

| Method | IR100 | | | | IR200 | IR50 | IR10 |
| | Head | Med | Tail | All | All | All | All |
| --- | --- | --- | --- | --- | --- | --- | --- |
| Training from scratch | | | | | | | |
| CE | 69.2 | 41.6 | 9.0 | 41.5 | 37.5 | 45.6 | 58.1 |
| CE+SAM | 72.7 | 41.8 | 7.0 | 42.2 | 38.9 | 46.8 | 59.7 |
| CE+ImbSAM | 68.5 | **46.0** | **9.6** | 43.0 | 38.7 | 47.8 | 60.1 |
| CE+CC-SAM | 70.1 | 44.2 | 9.0 | 42.7 | 39.1 | 47.4 | 60.0 |
| **CE+Focal-SAM** | **73.8** | 44.2 | 8.9 | **44.0** | **39.6** | **48.1** | **60.9** |
| LA (Menon et al., 2021) | 61.3 | 42.3 | 28.6 | 44.9 | 41.8 | 50.3 | 59.4 |
| LA+SAM | 63.1 | 52.2 | 32.2 | 50.0 | 45.5 | 52.8 | 62.6 |
| LA+ImbSAM | 57.4 | 51.1 | 31.0 | 47.3 | 43.4 | 52.2 | 62.4 |
| LA+CC-SAM | 63.7 | 51.9 | 32.3 | 50.1 | 45.6 | 53.0 | 63.0 |
| **LA+Focal-SAM** | **63.9** | **53.0** | **32.5** | **50.7** | **46.0** | **54.5** | **63.8** |
| Fine-tuning foundation model | | | | | | | |
| FFT | **88.2** | 79.3 | 66.1 | 78.5 | 76.3 | 81.2 | 85.5 |
| FFT+SAM | 87.9 | 82.5 | 70.8 | 80.9 | 77.7 | 83.4 | 86.8 |
| FFT+ImbSAM | 87.5 | 82.0 | 70.2 | 80.4 | 77.2 | 81.9 | 86.7 |
| FFT+CC-SAM | 87.8 | **82.9** | 70.9 | 81.0 | 78.2 | 83.5 | 87.0 |
| **FFT+Focal-SAM** | 88.1 | 82.8 | **72.4** | **81.6** | **79.0** | **83.9** | **87.3** |
| LIFT (Shi et al., 2024) | 85.3 | 81.1 | 79.2 | 82.0 | 79.6 | 82.8 | 85.0 |
| LIFT+SAM | 85.0 | 81.5 | **79.4** | 82.1 | 79.6 | 83.0 | 85.1 |
| LIFT+ImbSAM | 84.7 | **81.9** | 78.9 | 82.0 | 79.8 | 83.1 | 85.2 |
| LIFT+CC-SAM | 84.8 | 81.8 | 79.0 | 82.0 | 79.7 | 83.1 | 85.2 |
| **LIFT+Focal-SAM** | **85.4** | **81.9** | **79.4** | **82.4** | **80.0** | **83.2** | **85.4** |

each class group. For more details of the evaluation protocol, please refer to App.C.3.

**Implementation Details.** For CIFAR-LT datasets, we train ResNet models using ResNet-32 (He et al., 2016) as the backbone. For ImageNet-LT and iNaturalist datasets, we employ ResNet-50 (He et al., 2016). Training is conducted for 200 epochs. For fine-tuning foundation models, we follow the protocols outlined in LIFT (Shi et al., 2024). Specifically, we fine-tune the image encoder of CLIP (Radford et al., 2021) with a ViT-B/16 (Dosovitskiy et al., 2021) backbone. The training lasts for 20 epochs. For further implementation details, please refer to App.C.4.

### 5.2. Performance Comparison

Tab.2 summarizes the experimental results on the CIFAR-LT datasets with different imbalance ratios. From these results, we have the following observations: 1) Focal-SAM consistently performs better than SAM, ImbSAM, and CC-SAM across various loss functions. 2) Focal-SAM significantly outperforms ImbSAM on head classes, while maintaining

or surpassing ImbSAM on tail classes. Additionally, Focal-SAM generally outperforms CC-SAM on both head and tail classes, showing its ability to achieve a finer balance between head and tail classes performance.

Tab.3 presents results on the larger ImageNet-LT and iNaturalist datasets. Combining the baseline LA with Focal-SAM improves performance by approximately 1.9%~2.3% when training ResNet models. Similarly, pairing the baseline FFT or LIFT with Focal-SAM yields a performance gain of 0.3%~2.4% when fine-tuning foundation models, outperforming several competitors.

### 5.3. Long-tailed Domain Generalization

In Tab.4, we evaluate the model trained on the ImageNet-LT dataset across three OOD datasets. The results show the following: 1) SAM-based methods, when combined with FFT or LIFT, generally achieve more performance gain on OOD datasets than on the ID dataset (ImageNet-LT). This observation aligns with prior studies (Zhou et al., 2022; Khattak et al., 2023; Park et al., 2024), which suggest that

Table 3: Performance comparison on ImageNet-LT and iNaturalist. The results for methods marked with † are taken from the original paper. "-" indicates that the original paper didn't report the corresponding results.

| Method | ImageNet-LT | | | | iNaturalist | | | |
|---|---|---|---|---|---|---|---|---|
| | Head | Med | Tail | All | Head | Med | Tail | All |
| Training from scratch | | | | | | | | |
| CB (Cui et al., 2019) † | 39.6 | 32.7 | 16.8 | 33.2 | 53.4 | 54.8 | 53.2 | 54.0 |
| cRT (Kang et al., 2020) † | 61.8 | 46.2 | 27.3 | 49.6 | 69.0 | 66.0 | 63.2 | 65.2 |
| DiVE (He et al., 2021) † | 64.1 | 50.4 | 30.7 | 49.4 | 70.6 | 70.0 | 67.6 | 69.1 |
| DRO-LT (Samuel & Chechik, 2021) † | 64.0 | 49.8 | 33.1 | 53.5 | - | - | - | 69.7 |
| DisAlign (Zhang et al., 2021) † | 61.3 | 52.2 | 31.4 | 52.9 | 69.0 | 71.1 | 70.2 | 70.6 |
| WB (Alshammari et al., 2022) † | 62.5 | 50.4 | 41.5 | 53.9 | 71.2 | 70.4 | 69.7 | 70.2 |
| CC-SAM (Zhou et al., 2023b) † | 61.4 | 49.5 | 37.1 | 52.4 | 65.4 | 70.9 | 72.2 | 70.9 |
| LA (Menon et al., 2021) | 62.8 | 49.0 | 31.8 | 52.0 | **68.4** | 69.4 | 69.2 | 69.2 |
| LA+SAM | 63.1 | 51.7 | 33.1 | 53.6 | 68.3 | 70.8 | 71.9 | 71.0 |
| LA+ImbSAM | 62.6 | 50.3 | 32.6 | 52.6 | 68.0 | 70.2 | 70.2 | 69.9 |
| **LA+Focal-SAM** | **63.9** | **52.2** | **34.4** | **54.3** | **68.4** | **72.0** | **72.5** | **71.8** |
| Fine-tuning foundation model | | | | | | | | |
| Decoder (Wang et al., 2024c) † | - | - | - | 73.2 | - | - | - | 59.2 |
| LPT (Dong et al., 2023) † | - | - | - | - | - | - | 79.3 | 76.1 |
| FFT | 79.9 | 70.5 | 51.0 | 71.5 | **69.7** | 71.9 | 71.7 | 71.6 |
| FFT+SAM | **80.9** | 72.9 | 54.3 | 73.5 | 69.5 | 74.4 | **74.4** | 73.8 |
| FFT+ImbSAM | 80.6 | 72.6 | 52.2 | 72.9 | 68.5 | 73.4 | 73.8 | 73.1 |
| FFT+CC-SAM | 80.6 | 73.6 | 54.2 | 73.6 | 69.2 | 74.1 | 74.2 | 73.6 |
| **FFT+Focal-SAM** | 80.8 | **73.9** | **54.4** | **73.9** | 69.1 | **74.7** | 74.3 | **74.0** |
| LIFT (Shi et al., 2024) | 79.7 | 76.2 | 72.8 | 77.1 | **74.1** | 79.4 | 81.5 | 79.7 |
| LIFT+SAM | **79.9** | 76.4 | 72.7 | 77.2 | 73.5 | 79.7 | 81.6 | 79.8 |
| LIFT+ImbSAM | 79.8 | 76.4 | 72.5 | 77.2 | 73.2 | 79.5 | 81.4 | 79.6 |
| LIFT+CC-SAM | 79.8 | 76.4 | 73.3 | 77.3 | 74.0 | 79.4 | 81.5 | 79.7 |
| **LIFT+Focal-SAM** | 79.7 | **76.6** | **73.6** | **77.4** | 73.9 | **79.8** | **81.7** | **80.0** |

fine-tuning foundation models often perform well on target (ID) datasets but struggles with unseen (OOD) datasets. 2) Focal-SAM achieves a performance improvement of 0.5% to 4.3%, surpassing SAM, ImbSAM, and CC-SAM. This is because Focal-SAM effectively enhances the model's generalization ability by flattening the loss landscape, which mitigates performance issues on OOD test sets.

### 5.4. Training Speed of Focal-SAM

To assess the computational efficiency of Focal-SAM, we evaluate the training time per epoch across various long-tailed datasets, as shown in Tab.1. Focal-SAM requires about 50% more running time than SAM and has a similar running time to ImbSAM. Given that our method consistently outperforms SAM and ImbSAM, thus the computational cost is acceptable for the performance gain. Furthermore, Focal-SAM is significantly faster than CC-SAM while delivering better performance, aligning with our goal

of improving CC-SAM's efficiency.

### 5.5. Sharpness of Loss Landscape for Focal-SAM

To examine the impact of Focal-SAM on the loss landscape, Fig.2(d) and Fig.2(h) show the eigenvalue spectrum of Hessian for head and tail classes of models trained with Focal-SAM on CIFAR-10 LT using the VS loss function. Comparing Fig.2(e) and Fig.2(h), the trace $Tr(H)$ and the maximum eigenvalue $\lambda_{max}$ for tail classes in Focal-SAM are significantly lower than those in SAM. Similarly, Fig.2(b) and Fig.2(d) reveal that $Tr(H)$ and $\lambda_{max}$ for head classes in Focal-SAM are much smaller than those in ImbSAM. These results suggest that Focal-SAM achieves a fine-grained balance in the flatness between head and tail classes.

### 5.6. Ablation Study About $\gamma$ and $\lambda$

We analyze the influence of hyperparameters $\gamma$ and $\lambda$ to Focal-SAM on the CIFAR-LT datasets.

Table 4: Performance comparison for domain generalization. The source models are trained on the ImageNet-LT dataset and evaluated on out-of-distribution datasets, including ImageNet-Sketch, ImageNetV2, and ImageNet-C.

| Method | ImageNet-Sketch | | | | ImageNetV2 | | | | ImageNet-C | | | |
|---|---|---|---|---|---|---|---|---|---|---|---|---|
| | Head | Med | Tail | All | Head | Med | Tail | All | Head | Med | Tail | All |
| FFT | 42.9 | 35.5 | 21.4 | 36.4 | 70.1 | 60.2 | 45.2 | 62.0 | 50.3 | 41.4 | 26.1 | 42.8 |
| FFT+SAM | 44.9 | 39.3 | 26.1 | 39.6 | 71.2 | 62.6 | 48.0 | 63.9 | 52.5 | 44.6 | 29.3 | 45.6 |
| FFT+ImbSAM | 45.2 | 39.5 | 24.8 | 39.7 | 71.0 | 62.2 | 46.5 | 63.5 | 52.0 | 44.7 | 28.3 | 45.2 |
| FFT+CC-SAM | 45.0 | 41.0 | 26.8 | 40.6 | 71.3 | 63.2 | 48.4 | 64.3 | 52.0 | 45.1 | 29.4 | 45.6 |
| **FFT+Focal-SAM** | **45.5** | **41.2** | **27.3** | **41.0** | **71.8** | **63.6** | **48.8** | **64.8** | **52.6** | **45.5** | **29.8** | **46.1** |
| LIFT (Shi et al., 2024) | 46.4 | 43.3 | 45.7 | 44.8 | 70.4 | 65.9 | 64.7 | 67.5 | 52.6 | 48.7 | 47.3 | 50.0 |
| LIFT+SAM | **46.9** | 43.5 | 46.4 | 45.2 | **70.4** | 66.0 | 65.5 | 67.6 | 52.9 | 49.2 | 48.1 | 50.5 |
| LIFT+ImbSAM | 46.4 | 43.5 | 46.0 | 44.9 | 70.0 | 66.2 | 65.5 | 67.6 | 52.6 | 49.0 | 47.7 | 50.2 |
| LIFT+CC-SAM | 46.8 | 44.1 | 47.6 | 45.6 | **70.4** | 66.2 | 65.4 | 67.7 | 53.0 | 49.7 | 49.2 | 50.9 |
| **LIFT+Focal-SAM** | **46.9** | **44.7** | **49.4** | **46.2** | 70.0 | **66.8** | **66.9** | **68.0** | **53.1** | **49.9** | **49.8** | **51.1** |

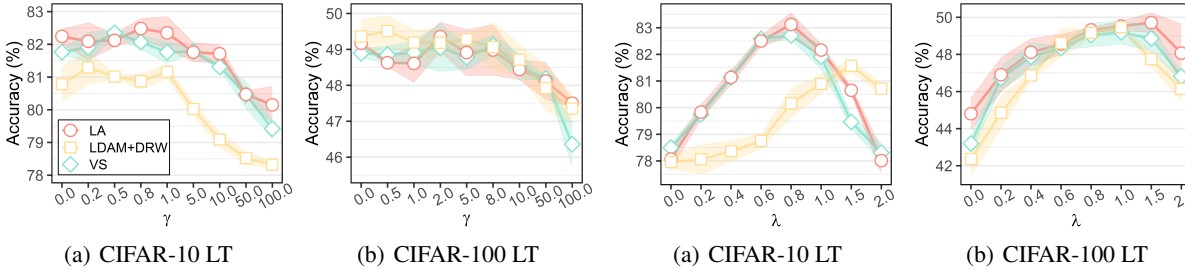

(a) CIFAR-10 LT  (b) CIFAR-100 LT

Figure 4: Ablation Study of Focal-SAM *w.r.t.* $\gamma$

(a) CIFAR-10 LT  (b) CIFAR-100 LT

Figure 5: Ablation Study of Focal-SAM *w.r.t.* $\lambda$

**Impact of $\gamma$:** Fig.4 explores the effect of $\gamma$. As $\gamma$ increases, performance initially improves, suggesting that assigning greater weight to the class-wise sharpness of tail classes benefits performance. However, a further increase in $\gamma$ leads to declining accuracy, indicating that assigning excessive weight to the class-wise sharpness of tail classes can harm performance.

**Impact of $\lambda$:** Fig.5 investigates the effect of $\lambda$. As $\lambda$ increases, accuracy initially improves but subsequently decreases. This indicates a trade-off between minimizing the training loss and minimizing the sharpness of the loss landscape.

## 6. Conclusion

This paper examines the limitations of ImbSAM and CC-SAM in long-tailed learning. ImbSAM excludes all head classes from SAM, often overemphasizing tail classes when combined with rebalancing methods. CC-SAM's per-class perturbation strategy provides fine-grained control over the loss landscape but is computationally costly. To address these issues, we propose Focal-SAM, a method that efficiently balances loss landscape flatness between head and tail classes. Additionally, we offer a theoretical analysis of

Focal-SAM's generalization ability, deriving a tighter bound. Extensive experiments validate Focal-SAM's effectiveness.

## Acknowledgements

This work was supported in part by the Fundamental Research Funds for the Central Universities, in part by the National Key R&D Program of China under Grant 2018AAA0102000, in part by National Natural Science Foundation of China: 62236008, 62441232, U21B2038, U23B2051, 62122075, 62025604, 62441619, 62206264 and 92370102, in part by Youth Innovation Promotion Association CAS, in part by the Strategic Priority Research Program of the Chinese Academy of Sciences, Grant No.XDB0680201, in part by the China National Postdoctoral Program for Innovative Talents under Grant BX20240384.

## Impact Statement

This paper presents work whose goal is to advance the field of Machine Learning. There are many potential societal consequences of our work, none which we feel must be specifically highlighted here.

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

# Appendix

## Contents

# A. Proof of Theorem

## A.1. Framework of the Proof

**Goal.** To bound the balanced loss $L_{\mathcal{D}_{bal}}(\boldsymbol{w})$ using our objective loss:

$$L_S^{FS}(\boldsymbol{w}) \triangleq \underbrace{[L_S(\boldsymbol{w}) - \lambda \cdot L_S^\gamma(\boldsymbol{w})]}_{(a)} + \underbrace{\lambda \cdot \max_{\|\boldsymbol{\epsilon}\|_2 \le \rho} L_S^\gamma(\boldsymbol{w} + \boldsymbol{\epsilon})}_{(b)} \tag{11}$$

**Framework of the proof.**

1. Essentially, the generalization bound describes how empirical values ($L_S^{FS}(\boldsymbol{w})$) deviate from the expected one ($L_{\mathcal{D}_{bal}}(\boldsymbol{w})$). To bound such deviations, Bernstein's inequality (Bernstein, 1924) and PAC-Bayesian theorem (Tolstikhin & Seldin, 2013) are convenient tools. Notice that these tools **require the empirical values to be sampled *i.i.d.* from the distribution on which the expectation is based**. Since the training set $S \sim D$, we first transform the distribution from $\mathcal{D}_{bal}$ to $D$ building on the work of Wang et al. (2023), *i.e.*,

$$L_{D_{bal}}(\boldsymbol{w}) \lesssim L_D(\boldsymbol{w}) \overset{\text{split into}}{=} \underbrace{[L_D(\boldsymbol{w}) - \lambda \cdot L_D^\gamma(\boldsymbol{w})]}_{(c)} + \underbrace{\lambda \cdot L_D^\gamma(\boldsymbol{w})}_{(d)} \tag{12}$$

2. Get $(c) \lesssim (a)$ via Bernstein's inequality (Lem.A.2).

3. Get $(d) \lesssim (b)$:

   - Bound $(d)$ using a intermediate value $\mathbb{E}_{\boldsymbol{\epsilon}}[L_D^\gamma(\boldsymbol{w} + \boldsymbol{\epsilon})]$ via Taylor expansion (Lem.A.4).
   - Bound $\mathbb{E}_{\boldsymbol{\epsilon}}[L_D^\gamma(\boldsymbol{w} + \boldsymbol{\epsilon})]$ by $(b)$ via PAC Bayesian bound (Lem.A.3).

Combine all, we get Thm.A.5 as follow:

$$L_{D_{bal}}(\boldsymbol{w}) \lesssim (c) + (d) \lesssim (a) + (b) = L_S^{FS}(\boldsymbol{w}) \tag{13}$$

## A.2. Proof of Lem.A.2

We begin by introducing Bernstein's inequality to prove the first part.

**Lemma A.1** (Bernstein's Inequality (Bernstein, 1924))**.** *Let $X_1, \cdots, X_n$ be i.i.d. random variables, $\mu = \mathbb{E}[X_1]$ and $\forall i, |X_i - \mu| \le b$. Let $\sigma^2 = Var(X_i)$. With probability at least $1 - \delta$,*

$$|\bar{X}_n - \mu| \le \sqrt{\frac{4\sigma^2 \log(\frac{2}{\delta})}{n}} + \frac{4b \log(\frac{2}{\delta})}{3n} \tag{14}$$

*where $\bar{X}_n = \frac{1}{n} \sum_{i=1}^n X_i$.*

Employing Lem.A.1, we can derive the following lemma to bound $L_{\mathcal{D}}(\boldsymbol{w}) - \lambda \cdot L_{\mathcal{D}}^\gamma(\boldsymbol{w})$ by $L_S(\boldsymbol{w}) - \lambda \cdot L_S^\gamma(\boldsymbol{w})$.

**Lemma A.2.** *Assume that $\forall(\boldsymbol{x}, y) \in \mathcal{D}, 0 \le \ell(\boldsymbol{w}; \boldsymbol{x}, y) \le B$. With probability $1 - \delta$ over the choice of the training set $S \sim \mathcal{D}$*

$$\Phi_{\mathcal{D}}^\lambda(\boldsymbol{w}) \le 2 \cdot \Phi_S^\lambda(\boldsymbol{w}) + \frac{40 \cdot (B + \lambda B') \cdot \log(\frac{2}{\delta})}{3n} \tag{15}$$

*where $B' \triangleq \sum_{i=1}^C (1 - \pi_i)^\gamma \pi_i B$, $\Phi_{\mathcal{D}}^\lambda(\boldsymbol{w}) \triangleq L_{\mathcal{D}}(\boldsymbol{w}) - \lambda \cdot L_{\mathcal{D}}^\gamma(\boldsymbol{w})$ and $\Phi_S^\lambda(\boldsymbol{w}) \triangleq L_S(\boldsymbol{w}) - \lambda \cdot L_S^\gamma(\boldsymbol{w})$.*

*Proof.* Since $\forall(\boldsymbol{x}, y) \in \mathcal{D}, \forall \boldsymbol{w} \in \mathcal{W}, \ell(\boldsymbol{w}; \boldsymbol{x}, y) \le B$, we have

$$0 \le L_S(\boldsymbol{w}) \le B, 0 \le L_{\mathcal{D}}(\boldsymbol{w}) \le B \tag{16}$$

and

$$0 \leq L_S^\gamma(\boldsymbol{w}) = \sum_{i=1}^{C}(1 - \pi_i)^\gamma L_S^i(\boldsymbol{w}) \leq \sum_{i=1}^{C}(1 - \pi_i)^\gamma \pi_i B \triangleq B'$$

$$0 \leq L_\mathcal{D}^\gamma(\boldsymbol{w}) = \mathbb{E}_S[L_S^\gamma(\boldsymbol{w})] \leq \sum_{i=1}^{C}(1 - \pi_i)^\gamma \pi_i B \triangleq B' \tag{17}$$

By the above two inequalities, we can obtain

$$|\Phi_\mathcal{D}^\lambda(\boldsymbol{w})| \leq B + \lambda B'$$
$$|\Phi_S^\lambda(\boldsymbol{w})| \leq B + \lambda B' \tag{18}$$

Thus, we have

$$|\Phi_S^\lambda(\boldsymbol{w}) - \Phi_\mathcal{D}^\lambda(\boldsymbol{w})| \leq 2 \cdot (B + \lambda B') \tag{19}$$

To simplify the analysis, we assume $\Phi_\mathcal{D}^\lambda(\boldsymbol{w}) \geq 0$. This assumption is reasonable because our experiments in Sec.5.6 typically show that the best value for $\lambda$ is slightly less than 1, where this assumption holds true. With this assumption, the variance of $\Phi_S^\lambda(\boldsymbol{w})$ can be bounded as:

$$\mathrm{Var}(\Phi_S^\lambda(\boldsymbol{w})) \leq \mathbb{E}[(\Phi_S^\lambda(\boldsymbol{w}))^2] \leq 2 \cdot (B + \lambda B') \cdot \Phi_\mathcal{D}^\lambda(\boldsymbol{w}) \tag{20}$$

Using Lem.A.1, with probability at least $1 - \delta$, we have

$$\begin{aligned}
\Phi_\mathcal{D}^\lambda(\boldsymbol{w}) &\leq \Phi_S^\lambda(\boldsymbol{w}) + \sqrt{\frac{8 \cdot (B + \lambda B') \cdot \Phi_\mathcal{D}^\lambda(\boldsymbol{w}) \cdot \log(\frac{2}{\delta})}{n}} \\
&\quad + \frac{8 \cdot (B + \lambda B') \cdot \log(\frac{2}{\delta})}{3n} \\
&\leq \Phi_S^\lambda(\boldsymbol{w}) + \frac{1}{2} \cdot \Phi_\mathcal{D}^\lambda(\boldsymbol{w}) + \frac{20 \cdot (B + \lambda B') \cdot \log(\frac{2}{\delta})}{3n}
\end{aligned} \tag{21}$$

where the last inequality leverages the property that for any positive numbers $a$ and $b$, $\sqrt{ab} \leq \frac{a}{2} + \frac{b}{2}$.

Reformulate the inequality, we can obtain that with probability at least $1 - \delta$,

$$\Phi_\mathcal{D}^\lambda(\boldsymbol{w}) \leq 2 \cdot \Phi_S^\lambda(\boldsymbol{w}) + \frac{40 \cdot (B + \lambda B') \cdot \log(\frac{2}{\delta})}{3n} \tag{22}$$

$\square$

### A.3. Proof of Lem.A.3 and Lem.A.4

The following lemmas utilize the PAC-Bayesian theorem to prove the second part. We first derive an intermediate result in the following lemma.

**Lemma A.3.** *Assume that $\forall (\boldsymbol{x}, y) \in \mathcal{D}, 0 \leq \ell(\boldsymbol{w}; \boldsymbol{x}, y) \leq B$. Then, for any $\rho > 0$ and any distribution $\mathcal{D}$, with probability $1 - \delta$ over the choice of the training set $S \sim \mathcal{D}$*

$$\begin{aligned}
\mathbb{E}_{\epsilon_i \sim \mathcal{N}(0, \sigma_Q)}[L_\mathcal{D}^\gamma(\boldsymbol{w} + \boldsymbol{\epsilon})] &\leq \max_{\|\boldsymbol{\epsilon}\|_2 \leq \rho} 2L_S^\gamma(\boldsymbol{w} + \boldsymbol{\epsilon}) \\
&\quad + \frac{2 + 2B' + 2k \log\left(1 + \frac{\|\boldsymbol{w}\|_2^2}{k\rho^2}\right) + 4k \log\left(\sqrt{k} + \sqrt{2\ln n}\right) + 4\log\frac{\pi^2\sqrt{n}(nB'+1)^2}{3\delta}}{n}
\end{aligned} \tag{23}$$

*where $n = |S|$, $k$ is the number of parameters, $B' \triangleq \sum_{i=1}^{C}(1 - \pi_i)^\gamma \pi_i B$ and $\sigma_Q \triangleq \frac{\rho}{\sqrt{k} + \sqrt{2\ln(n)}}$.*

*Proof.* Inspired by the proof technique in SAM (Foret et al., 2021), we provide the following proof.

Since $\forall(\boldsymbol{x}, y) \in \mathcal{D}, \forall \boldsymbol{w} \in \mathcal{W}, \ell(\boldsymbol{w}; \boldsymbol{x}, y) \leq B$, we have:

$$L_S^\gamma(\boldsymbol{w}) = \sum_{i=1}^C (1-\pi_i)^\gamma L_S^i(\boldsymbol{w}) \leq \sum_{i=1}^C (1-\pi_i)^\gamma \pi_i B = B' \tag{24}$$

$$L_\mathcal{D}^\gamma(\boldsymbol{w}) = \mathbb{E}_S[L_S^\gamma(\boldsymbol{w})] \leq \sum_{i=1}^C (1-\pi_i)^\gamma \pi_i B = B' \tag{25}$$

Thereby, the right-hand side of the bound in the theorem is lower bounded by $\frac{k}{n} \log(1 + \frac{\|\boldsymbol{w}\|_2^2}{k\rho^2})$ which is greater than $B'$ when $\|\boldsymbol{w}\|_2^2 > k\rho^2[\exp(nB'/k) - 1]$ and in this case the inequality holds trivially. Thereby, we only consider the case when $\|\boldsymbol{w}\|_2^2 \leq k\rho^2[\exp(nB'/k) - 1]$ in the rest of the proof.

Using PAC-Bayesian generalization bound in (Tolstikhin & Seldin, 2013), for any fixed prior $\mathcal{P}$ over parameters, with probability $1 - \delta$ over training set $S$, for any posterior $\mathcal{Q}$ over parameters, the following generalization bound holds:

$$\begin{aligned} \mathbb{E}_{\boldsymbol{w} \sim \mathcal{Q}}[L_\mathcal{D}^\gamma(\boldsymbol{w})] &\leq \mathbb{E}_{\boldsymbol{w} \sim \mathcal{Q}}[L_S^\gamma(\boldsymbol{w})] + \sqrt{2\mathbb{E}_{\boldsymbol{w} \sim \mathcal{Q}}[L_S^\gamma(\boldsymbol{w})] \frac{KL(\mathcal{Q}\|\mathcal{P}) + \log \frac{2\sqrt{n}}{\delta}}{n}} \\ &\quad + 2\frac{KL(\mathcal{Q}\|\mathcal{P}) + \log \frac{2\sqrt{n}}{\delta}}{n} \\ &\leq 2\mathbb{E}_{\boldsymbol{w} \sim \mathcal{Q}}[L_S^\gamma(\boldsymbol{w})] + 4\frac{KL(\mathcal{Q}\|\mathcal{P}) + \log \frac{2\sqrt{n}}{\delta}}{n} \end{aligned} \tag{26}$$

where the last inequality leverages the property that for any positive numbers $a$ and $b$, $\sqrt{ab} \leq a + b$.

Following SAM (Foret et al., 2021), we assume $\mathcal{P} = \mathcal{N}(\boldsymbol{\mu}_P, \sigma_P^2 \boldsymbol{I})$ and $\mathcal{Q} = \mathcal{N}(\boldsymbol{\mu}_Q, \sigma_Q^2 \boldsymbol{I})$, then the KL divergence can be written as:

$$KL(\mathcal{Q}\|\mathcal{P}) = \frac{1}{2}\left[\frac{k\sigma_Q^2 + \|\boldsymbol{\mu}_P - \boldsymbol{\mu}_Q\|_2^2}{\sigma_P^2} - k + k \log\left(\frac{\sigma_P^2}{\sigma_Q^2}\right)\right] \tag{27}$$

Let $T = \{c\exp((1-j)/k)|j \in \mathbb{N}\}$ be the predefined set of values for $\sigma_P^2$. If for any $j \in \mathbb{N}$, the bounds holds with probability $1 - \delta_j$ with $\delta_j = \frac{6\delta}{\pi^2 j^2}$, then by the union bound, all above bounds hold simultaneously with probability $1 - \sum_{j=1}^\infty \frac{6\delta}{\pi^2 j^2} = 1 - \delta$.

Let $\sigma_Q = \frac{\rho}{\sqrt{k} + \sqrt{2\ln(n)}}, \boldsymbol{\mu}_Q = \boldsymbol{w}$ and $\boldsymbol{\mu}_P = \boldsymbol{0}$. We have:

$$\sigma_Q^2 + \frac{\|\boldsymbol{\mu}_P - \boldsymbol{\mu}_Q\|_2^2}{k} \leq \rho^2 + \frac{\|\boldsymbol{w}\|_2^2}{k} \leq \rho^2 \exp(\frac{nB'}{k}) \tag{28}$$

Let $j = \lfloor 1 - k \log((\rho^2 + \|\boldsymbol{w}\|_2^2/k)/c) \rfloor$. We can ensure $j \in \mathbb{N}$ by setting $c = \rho^2 \exp(nB'/k)$. For $\sigma_P^2 = c\exp((1-j)/k)$, we have:

$$\rho^2 + \frac{\|\boldsymbol{w}\|_2^2}{k} \leq \sigma_P^2 \leq \exp(\frac{1}{k})(\rho^2 + \frac{\|\boldsymbol{w}\|_2^2}{k}) \tag{29}$$

Building on Eq.(28) and Eq.(29), we can obtain an upper bound for the KL divergence:

$$KL(\mathcal{Q}\|\mathcal{P}) = \frac{1}{2}\left[\frac{k\sigma_Q^2 + \|\boldsymbol{\mu}_P - \boldsymbol{\mu}_Q\|_2^2}{\sigma_P^2} - k + k\log\left(\frac{\sigma_P^2}{\sigma_Q^2}\right)\right] \tag{30}$$

$$\leq \frac{1}{2}\left[\frac{k(\rho^2 + \frac{\|\boldsymbol{w}\|_2^2}{k})}{\rho^2 + \frac{\|\boldsymbol{w}\|_2^2}{k}} - k + k\log\left(\frac{\exp(\frac{1}{k})(\rho^2 + \frac{\|\boldsymbol{w}\|_2^2}{k})}{\sigma_Q^2}\right)\right] \tag{31}$$

$$= \frac{1}{2}\left[k\log\left(\frac{\exp(\frac{1}{k})(\rho^2 + \frac{\|\boldsymbol{w}\|_2^2}{k})}{\sigma_Q^2}\right)\right] \tag{32}$$

$$= \frac{1}{2}\left[k\log\left(\frac{\exp(\frac{1}{k})(\rho^2 + \frac{\|\boldsymbol{w}\|_2^2}{k})(\sqrt{k} + \sqrt{2\ln n})^2}{\rho^2}\right)\right] \tag{33}$$

$$= \frac{1}{2}\left[1 + k\log\left(1 + \frac{\|\boldsymbol{w}\|_2^2}{k\rho^2}\right) + 2k\log\left(\sqrt{k} + \sqrt{2\ln n}\right)\right] \tag{34}$$

Given the bound that corresponds to $j$ holds with probability $1 - \delta_j$ for $\delta_j = \frac{6\delta}{\pi^2 j^2}$, the log term can be bounded as:

$$\log\frac{2\sqrt{n}}{\delta_j} = \log\frac{2\sqrt{n}}{\delta} + \log\frac{\pi^2 j^2}{6} \tag{35}$$

$$\leq \log\frac{2\sqrt{n}}{\delta} + \log\frac{\pi^2(1 + \log(c/\rho^2))^2}{6} \tag{36}$$

$$\leq \log\frac{2\sqrt{n}}{\delta} + \log\frac{\pi^2(1 + k\log(\exp(nB'/k)))^2}{6} \tag{37}$$

$$= \log\frac{2\sqrt{n}}{\delta} + \log\frac{\pi^2(1 + nB')^2}{6} \tag{38}$$

$$= \log\frac{\pi^2\sqrt{n}(1 + nB')^2}{3\delta} \tag{39}$$

where the first inequality is derived from the fact that $j \leq 1 + k\log(c/(\rho^2 + \|\boldsymbol{w}\|_2^2/k)) \leq 1 + k\log(c/\rho^2)$.

Therefore, the generalization bound can be written as:

$$\mathbb{E}_{\epsilon_i \sim \mathcal{N}(0,\sigma_Q)}[L_{\mathcal{D}}^\gamma(\boldsymbol{w} + \boldsymbol{\epsilon})] \leq 2\mathbb{E}_{\epsilon_i \sim \mathcal{N}(0,\sigma_Q)}[L_S^\gamma(\boldsymbol{w} + \boldsymbol{\epsilon})]$$
$$+ \frac{2 + 2k\log\left(1 + \frac{\|\boldsymbol{w}\|_2^2}{k\rho^2}\right) + 4k\log\left(\sqrt{k} + \sqrt{2\ln n}\right) + 4\log\frac{\pi^2\sqrt{n}(1+nB')^2}{3\delta}}{n} \tag{40}$$

Since $\|\boldsymbol{\epsilon}\|_2^2$ has chi-square distribution, for any positive $t$, we have:

$$P(\|\boldsymbol{\epsilon}\|_2^2 - k\sigma_Q^2 \geq 2\sigma_Q^2\sqrt{kt} + 2t\sigma_Q^2) \leq \exp(-t) \tag{41}$$

Therefore, with probability $1 - 1/n$, we have:

$$\|\boldsymbol{\epsilon}\|_2^2 \leq \sigma_Q^2\left[k + 2\sqrt{k\ln(n)} + 2\ln(n)\right] \tag{42}$$

$$\leq \sigma_Q^2\left[\sqrt{k} + \sqrt{2\ln(n)}\right]^2 \tag{43}$$

$$\leq \rho^2 \tag{44}$$

Therefore, we have:

$$\mathbb{E}_{\epsilon_i \sim \mathcal{N}(0,\sigma_Q)}[L_\mathcal{D}^\gamma(\boldsymbol{w} + \boldsymbol{\epsilon})] \leq 2(1 - 1/n) \max_{\|\boldsymbol{\epsilon}\|_2 \leq \rho} L_S^\gamma(\boldsymbol{w} + \boldsymbol{\epsilon}) + \frac{2B'}{n}$$

$$+ \frac{2 + 2k \log\left(1 + \frac{\|\boldsymbol{w}\|_2^2}{k\rho^2}\right) + 4k \log\left(\sqrt{k} + \sqrt{2\ln n}\right) + 4\log \frac{\pi^2 \sqrt{n}(nB'+1)^2}{3\delta}}{n}$$

$$\leq \max_{\|\boldsymbol{\epsilon}\|_2 \leq \rho} 2L_S^\gamma(\boldsymbol{w} + \boldsymbol{\epsilon})$$

$$+ \frac{2 + 2B' + 2k \log\left(1 + \frac{\|\boldsymbol{w}\|_2^2}{k\rho^2}\right) + 4k \log\left(\sqrt{k} + \sqrt{2\ln n}\right) + 4\log \frac{\pi^2 \sqrt{n}(nB'+1)^2}{3\delta}}{n} \tag{45}$$

$\square$

Combining the above lemma with the Taylor expansion, we can derive the following lemma to bound $\lambda \cdot L_\mathcal{D}^\gamma(\boldsymbol{w})$ by $\lambda \cdot \max_{\|\boldsymbol{\epsilon}\|_2 \leq \rho} L_S^\gamma(\boldsymbol{w} + \boldsymbol{\epsilon})$.

**Lemma A.4.** *Assume that $\forall (\boldsymbol{x}, y) \in \mathcal{D}, 0 \leq \ell(\boldsymbol{w}; \boldsymbol{x}, y) \leq B$. Then, for any $\rho > 0$ and any distribution $\mathcal{D}$, with probability $1 - \delta$ over the choice of the training set $S \sim \mathcal{D}$*

$$L_\mathcal{D}^\gamma(\boldsymbol{w}) \leq \max_{\|\boldsymbol{\epsilon}\|_2 \leq \rho} 2L_S^\gamma(\boldsymbol{w} + \boldsymbol{\epsilon}) - \frac{\rho^2}{2(\sqrt{k} + \sqrt{2\ln(n)})^2} \cdot tr(H(\boldsymbol{w})) - o\left(\frac{k\rho^2}{(\sqrt{k} + \sqrt{2\ln(n)})^2}\right)$$

$$+ \frac{2 + 2B' + 2k \log\left(1 + \frac{\|\boldsymbol{w}\|_2^2}{k\rho^2}\right) + 4k \log\left(\sqrt{k} + \sqrt{2\ln n}\right) + 4\log \frac{\pi^2 \sqrt{n}(nB'+1)^2}{3\delta}}{n} \tag{46}$$

*where $n = |S|$, $k$ is the number of parameters, $B' \triangleq \sum_{i=1}^C (1 - \pi_i)^\gamma \pi_i B$, $H(\boldsymbol{w})$ represents the Hessian matrix of $L_\mathcal{D}^\gamma(\boldsymbol{w})$ at point $\boldsymbol{w}$ and $tr(\cdot)$ represent the matrix trace.*

*Proof.* By expanding $\mathbb{E}_{\epsilon_i \sim \mathcal{N}(0,\sigma_Q)}[L_\mathcal{D}^\gamma(\boldsymbol{w} + \boldsymbol{\epsilon})]$ around $\boldsymbol{w}$ using a second-order Taylor Series expansion, we can obtain

$$\mathbb{E}_{\epsilon_i \sim \mathcal{N}(0,\sigma_Q)}[L_\mathcal{D}^\gamma(\boldsymbol{w} + \boldsymbol{\epsilon})] = \mathbb{E}_{\epsilon_i \sim \mathcal{N}(0,\sigma_Q)}[L_\mathcal{D}^\gamma(\boldsymbol{w}) + \boldsymbol{\epsilon}^T \nabla L_\mathcal{D}^\gamma(\boldsymbol{w}) + \frac{1}{2}\boldsymbol{\epsilon}^T H(\boldsymbol{w})\boldsymbol{\epsilon} + o(\|\boldsymbol{\epsilon}\|_2^2)]$$

$$= L_\mathcal{D}^\gamma(\boldsymbol{w}) + \frac{1}{2}\mathbb{E}_{\epsilon_i \sim \mathcal{N}(0,\sigma_Q)}[\boldsymbol{\epsilon}^T H(\boldsymbol{w})\boldsymbol{\epsilon}] + \mathbb{E}_{\epsilon_i \sim N(0,\sigma_Q)}[o(\|\boldsymbol{\epsilon}\|_2^2)] \tag{47}$$

where $\sigma_Q \triangleq \frac{\rho}{\sqrt{k} + \sqrt{2\ln(n)}}$ and $H(\boldsymbol{w})$ represents the Hessian matrix of $L_\mathcal{D}^\gamma(\boldsymbol{w})$ at point $\boldsymbol{w}$.

Thereby, we have:

$$\mathbb{E}_{\epsilon_i \sim \mathcal{N}(0,\sigma_Q)}[L_\mathcal{D}^\gamma(\boldsymbol{w} + \boldsymbol{\epsilon})] = L_\mathcal{D}^\gamma(\boldsymbol{w}) + \frac{1}{2}\mathbb{E}_{\epsilon_i \sim \mathcal{N}(0,\sigma_Q)}[\boldsymbol{\epsilon}^T H(\boldsymbol{w})\boldsymbol{\epsilon}] + \mathbb{E}_{\epsilon_i \sim N(0,\sigma_Q)}[o(\|\boldsymbol{\epsilon}\|_2^2)]$$

$$= L_\mathcal{D}^\gamma(\boldsymbol{w}) + \frac{tr(H(\boldsymbol{w}))}{2} \cdot \mathbb{E}_{\boldsymbol{\epsilon}_1 \sim N(0,\sigma_Q)}[\epsilon_1^2] + o(k \cdot \mathbb{E}_{\boldsymbol{\epsilon}_1 \sim N(0,\sigma_Q)}[\epsilon_1^2]) \tag{48}$$

$$= L_\mathcal{D}^\gamma(\boldsymbol{w}) + \frac{\rho^2}{2(\sqrt{k} + \sqrt{2\ln(n)})^2} \cdot tr(H(\boldsymbol{w})) + o\left(\frac{k\rho^2}{(\sqrt{k} + \sqrt{2\ln(n)})^2}\right)$$

Combining Eq.(48) with Lem.A.3, with probability $1 - \delta$, we have

$$L_\mathcal{D}^\gamma(\boldsymbol{w}) \leq \max_{\|\boldsymbol{\epsilon}\|_2 \leq \rho} 2L_S^\gamma(\boldsymbol{w} + \boldsymbol{\epsilon}) - \frac{\rho^2}{2(\sqrt{k} + \sqrt{2\ln(n)})^2} \cdot tr(H(\boldsymbol{w})) - o\left(\frac{k\rho^2}{(\sqrt{k} + \sqrt{2\ln(n)})^2}\right)$$

$$+ \frac{2 + 2B' + 2k \log\left(1 + \frac{\|\boldsymbol{w}\|_2^2}{k\rho^2}\right) + 4k \log\left(\sqrt{k} + \sqrt{2\ln n}\right) + 4\log \frac{\pi^2 \sqrt{n}(nB'+1)^2}{3\delta}}{n} \tag{49}$$

$\square$

## A.4. Proof of Thm.4.1

Combining the above two parts, we can finally derive the following theorem.

**Theorem A.5** (Restate of Thm.4.1)**.** *Assume that* $\forall(\boldsymbol{x}, y) \in \mathcal{D}, 0 \leq \ell(\boldsymbol{w}; \boldsymbol{x}, y) \leq B$. *For any* $\rho > 0$, *any uniform distribution* $\mathcal{D}_{bal}$ *and any distribution* $\mathcal{D}$, *with probability* $1 - \delta$ *over the choice of the training set* $S \sim \mathcal{D}$,

$$
\begin{aligned}
L_{\mathcal{D}_{bal}}(\boldsymbol{w}) \leq {}& \frac{2L_S^{FS}(\boldsymbol{w})}{C\pi_C} + \frac{40 \cdot (B + \lambda B') \cdot \log(\frac{4}{\delta})}{3n \cdot C\pi_C} - \frac{\lambda\rho^2}{2(\sqrt{k} + \sqrt{2\ln(n)})^2 \cdot C\pi_C} \cdot tr(H(\boldsymbol{w})) \\
& + \lambda \cdot \frac{2 + 2B' + 2k\log\left(1 + \frac{\|\boldsymbol{w}\|_2^2}{k\rho^2}\right) + 4k\log\left(\sqrt{k} + \sqrt{2\ln n}\right) + 4\log\frac{2\pi^2\sqrt{n}(nB'+1)^2}{3\delta}}{n \cdot C\pi_C} \\
& - o(\frac{\lambda k\rho^2}{(\sqrt{k} + \sqrt{2\ln(n)})^2 \cdot C\pi_C})
\end{aligned}
\tag{50}
$$

*where* $n = |S|$, $k$ *is the number of parameters,* $B' \triangleq \sum_{i=1}^{C}(1 - \pi_i)^\gamma \pi_i B$, $H(\boldsymbol{w})$ *represents the Hessian matrix of* $L_{\mathcal{D}}^\gamma(\boldsymbol{w})$ *at point* $\boldsymbol{w}$ *and* $tr(\cdot)$ *represent the matrix trace.*

*Proof.* Combining Lem.A.2 and Lem.A.4 and using union bound, with probability at least $1 - \delta$, we have

$$
\begin{aligned}
L_{\mathcal{D}}(\boldsymbol{w}) \leq {}& 2L_S^{FS}(\boldsymbol{w}) + \frac{40 \cdot (B + \lambda B') \cdot \log(\frac{4}{\delta})}{3n} - \frac{\lambda\rho^2}{2(\sqrt{k} + \sqrt{2\ln(n)})^2} \cdot tr(H(\boldsymbol{w})) \\
& + \lambda \cdot \frac{2 + 2B' + 2k\log\left(1 + \frac{\|\boldsymbol{w}\|_2^2}{k\rho^2}\right) + 4k\log\left(\sqrt{k} + \sqrt{2\ln n}\right) + 4\log\frac{2\pi^2\sqrt{n}(nB'+1)^2}{3\delta}}{n} \\
& - o(\frac{\lambda k\rho^2}{(\sqrt{k} + \sqrt{2\ln(n)})^2})
\end{aligned}
\tag{51}
$$

We further recognize that:

$$
L_{\mathcal{D}}(\boldsymbol{w}) = \sum_{i=1}^{C} \pi_i L_{\mathcal{D}_i}(\boldsymbol{w}) \geq \sum_{i=1}^{C} \pi_C L_{\mathcal{D}_i}(\boldsymbol{w}) = C\pi_C \cdot L_{\mathcal{D}_{bal}}(\boldsymbol{w})
\tag{52}
$$

Substituting Eq.(52) into Eq.(51) leads to Thm.A.5.

$\square$

# B. Analysis of Backpropagation Requirements for SAM and ImbSAM

## B.1. Backpropagation Requirements for SAM

SAM aims to find flatter minima, ensuring the entire neighborhood around the model parameters has consistently low training loss. The objective loss function is defined as:

$$
L_S^{SAM}(\boldsymbol{w}) \triangleq \max_{\|\boldsymbol{\epsilon}\|_2 \leq \rho} L_S(\boldsymbol{w} + \boldsymbol{\epsilon})
\tag{53}
$$

The optimal perturbation $\hat{\boldsymbol{\epsilon}}_{SAM}(\boldsymbol{w})$ for the inner maximization problem is estimated as follow:

$$
\hat{\boldsymbol{\epsilon}}_{SAM}(\boldsymbol{w}) \approx \rho \frac{\nabla_{\boldsymbol{w}} L_S(\boldsymbol{w})}{\|\nabla_{\boldsymbol{w}} L_S(\boldsymbol{w})\|_2}
\tag{54}
$$

Thus, the gradient of $L_S^{SAM}(\boldsymbol{w})$ can be approximated as:

$$
\nabla_{\boldsymbol{w}} L_S^{SAM}(\boldsymbol{w}) \approx \nabla_{\boldsymbol{w}} L_S(\boldsymbol{w})\big|_{\boldsymbol{w} + \hat{\boldsymbol{\epsilon}}(\boldsymbol{w})}
\tag{55}
$$

To update parameters once using SAM, **two** backpropagations are required: one for $\nabla_{\boldsymbol{w}} L_S(\boldsymbol{w})$, and another for $\nabla_{\boldsymbol{w}} L_S(\boldsymbol{w})\big|_{\boldsymbol{w} + \hat{\boldsymbol{\epsilon}}(\boldsymbol{w})}$

## B.2. Backpropagation Requirements for ImbSAM

ImbSAM divides classes into head and tail groups, denoted as $\mathcal{H}$ and $\mathcal{T}$, and applies SAM only to the tail group. Its objective function is:

$$L_S^{IS}(\boldsymbol{w}) \triangleq L_S^{\mathcal{H}}(\boldsymbol{w}) + \max_{\|\boldsymbol{\epsilon}\|_2 \leq \rho} L_S^{\mathcal{T}}(\boldsymbol{w} + \boldsymbol{\epsilon}) \tag{56}$$

The optimal perturbation $\hat{\boldsymbol{\epsilon}}_{IS}(\boldsymbol{w})$ for the inner maximization problem is estimated as follow:

$$\hat{\boldsymbol{\epsilon}}_{IS}(\boldsymbol{w}) \approx \rho \frac{\nabla_{\boldsymbol{w}} L_S^{\mathcal{T}}(\boldsymbol{w})}{\|\nabla_{\boldsymbol{w}} L_S^{\mathcal{T}}(\boldsymbol{w})\|_2} \tag{57}$$

Thus, the gradient of $L_S^{IS}(\boldsymbol{w})$ can be approximated as:

$$\nabla_{\boldsymbol{w}} L_S^{IS}(\boldsymbol{w}) \approx \nabla_{\boldsymbol{w}} L_S^{\mathcal{H}}(\boldsymbol{w}) + \nabla_{\boldsymbol{w}} L_S^{\mathcal{T}}(\boldsymbol{w})\big|_{\boldsymbol{w} + \hat{\boldsymbol{\epsilon}}(\boldsymbol{w})} \tag{58}$$

To update parameters once using ImbSAM, **three** backpropagations are required: one for $\nabla_{\boldsymbol{w}} L_S^{\mathcal{T}}(\boldsymbol{w})$, one for $\nabla_{\boldsymbol{w}} L_S^{\mathcal{H}}(\boldsymbol{w})$, and another for $\nabla_{\boldsymbol{w}} L_S^{\mathcal{T}}(\boldsymbol{w})\big|_{\boldsymbol{w} + \hat{\boldsymbol{\epsilon}}(\boldsymbol{w})}$

# C. More Experiment Protocols

## C.1. Datasets

For long-tailed recognition tasks, we conduct experiments on four widely used long-tailed datasets: CIFAR-10 LT, CIFAR-100 LT, ImageNet-LT, and iNaturalist. For long-tailed domain generalization tasks, we train the model on ImageNet-LT and evaluate it on three OOD datasets: ImageNet-Sketch, ImageNetV2, and ImageNet-C. Below is a detailed description of these datasets:

- **CIFAR-100 LT and CIFAR-10 LT** (Cao et al., 2019). The original CIFAR-100 (Krizhevsky & Hinton, 2009) and CIFAR-10 (Krizhevsky & Hinton, 2009) datasets contain 50,000 training images and 10,000 testing images for 100 and 10 classes, respectively. We utilize their various long-tailed versions with different imbalance ratios of {100, 50, 10}.

- **ImageNet-LT** (Liu et al., 2019). The ImageNet-LT dataset is derived from the ImageNet (Deng et al., 2009) dataset according to a Pareto distribution, containing 1000 categories. The dataset includes 115,846 training images and 50,000 test images. The dataset has an imbalance ratio of 256.

- **iNaturalist** (Horn et al., 2018). The iNaturalist dataset is a real-world large-scale dataset, consisting of 8142 categories. The training set contains approximately 430,000 images, while the test set contains about 24,000 images. The dataset's imbalance ratio is 500.

- **ImageNet-Sketch** (Wang et al., 2019a). The ImageNet-Sketch dataset is an OOD test set derived from the ImageNet (Deng et al., 2009) dataset, comprising 50,000 images across 1000 classes. Each image is a sketch, introducing a domain shift relative to ImageNet.

- **ImageNetV2** (Recht et al., 2019). The ImageNetV2 dataset consists of 10,000 images spanning the same 1000 classes as ImageNet. The images are sourced differently from the original ImageNet (Deng et al., 2009), resulting in a slight domain shift.

- **ImageNet-C** (Hendrycks & Dietterich, 2019). The ImageNet-C dataset includes the same 1000 classes as ImageNet (Deng et al., 2009) but features corrupted versions of the original validation set. Each image undergoes one of 15 corruption types at 5 severity levels, resulting in 75 dataset variations.

## C.2. Competitors

When training ResNet models from scratch, we evaluate serveral competitive methods on different datasets. For the CIFAR-LT dataset, we assess multiple loss functions, including CE loss, LDAM+DRW (Cao et al., 2019), LA loss (Menon et al., 2021), and VS loss (Kini et al., 2021). These methods are further combined with SAM (Foret et al., 2021), ImbSAM (Zhou et al., 2023a), and CC-SAM (Zhou et al., 2023b) as baseline comparisons. For the ImageNet-LT and iNaturalist datasets, we

employ a range of representative methods, including CB (Cui et al., 2019) for class re-balancing, cRT (Kang et al., 2020) for decoupled training, DiVE (He et al., 2021) for transfer learning, DRO-LT (Samuel & Chechik, 2021) for representation learning, DisAlign (Zhang et al., 2021) for class re-balancing, and WB (Alshammari et al., 2022) for regularization. When fine-tuning the foundation model CLIP (Radford et al., 2021), we use Decoder (Wang et al., 2024c) and LPT (Dong et al., 2023) as baselines. We also evaluate both fully fine-tuning the models with LA loss (denoted as FFT), and parameter-efficient fine-tuning using the LIFT method (Shi et al., 2024), as well as their performance when combined with different SAM variants.

## C.3. Evaluation Protocol

For long-tailed recognition tasks, we assess model performance using top-1 accuracy on balanced test sets. This ensures all classes contribute equally to the evaluation. To provide a more detailed analysis, we follow the approach in (Zhong et al., 2021; Liu et al., 2019) by splitting the classes into three subsets: Head, Medium, and Tail. Accuracy is then reported for each subset individually. For CIFAR-10 LT (IR = 100), the Head classes contain more than 1000 samples, the Medium classes have 200∼1000 samples, and the Tail classes have less than 200 samples. For CIFAR-100 LT (IR = 100), ImageNet-LT, and iNaturalist, the Head classes contain more than 100 samples, the Medium classes have 20∼100 samples, and the Tail classes have less than 20 samples. Prior arts (Zhou et al., 2022; Khattak et al., 2023; Park et al., 2024) have demonstrated that fine-tuning CLIP (Radford et al., 2021) often performs well on the target domain but struggles with domain shifts. Therefore, when fine-tuning the foundation models, we also assess model performance on OOD test sets, referred to as long-tailed domain generalization tasks. Specifically, models are trained on the ImageNet-LT dataset and evaluated on out-of-distribution datasets, including ImageNet-Sketch (Wang et al., 2019a), ImageNetV2 (Recht et al., 2019), and ImageNet-C (Hendrycks & Dietterich, 2019). We evaluate model performance on these OOD balanced test sets, including top-1 accuracy and accuracy for each class subset.

## C.4. Implementation Details

We follow the procedures described below to train ResNet models from scratch. For the CIFAR-LT datasets, we use ResNet-32 (He et al., 2016) as the backbone. We employ stochastic gradient descent (SGD) as the base optimizer, with an initial learning rate of 0.1, a batch size of 64, and a momentum of 0.9. Training spans 200 epochs, using a cosine annealing scheduler to reduce the learning rate from 0.1 to 0 gradually. For the larger-scale ImageNet-LT and iNaturalist datasets, we employ ResNet-50 (He et al., 2016) as the backbone. SGD is again used as the base optimizer with a momentum of 0.9. For ImageNet-LT, the initial learning rate is set to 0.1, with a batch size of 256, while for iNaturalist, the initial learning rate is 0.2, and the batch size is increased to 512. Training for these datasets also lasts 200 epochs with a cosine annealing scheduler. Additionally, We employ a step scheduler for $\rho$, following the approach of Rangwani et al. (2022). This scheduler initializes $\rho$ until the 160th epoch and then increases its value towards the end of training.

For fine-tuning foundation models, we follow the protocols outlined in LIFT (Shi et al., 2024). A cosine classifier is added after the image encoder of CLIP (Radford et al., 2021), with its weights initialized using the text encoder, which is then discarded. We fine-tune the image encoder of CLIP with a ViT-B/16 (Dosovitskiy et al., 2021) backbone. Stochastic gradient descent (SGD) is used as the base optimizer, with a batch size of 128 and momentum of 0.9. The initial learning rate is 0.01 for parameter-efficient fine-tuning and 0.001 for full fine-tuning. Unlike LIFT (Shi et al., 2024), all models in our experiments are fine-tuned for 20 epochs across datasets and methods. In LIFT, models are trained for 10 epochs on the CIFAR-LT and the ImageNet-LT datasets, and 20 epochs on the iNaturalist dataset. We extend the training to 20 epochs because the models do not fully converge under the original settings.

## C.5. Experimental Hardware Setup

All the experiments are conducted on Ubuntu servers equipped with Nvidia(R) RTX 3090 GPUs and RTX 4090 GPUs. Fine-tuning the foundation models is performed using a single GPU for all datasets. The number of GPUs used for training the ResNet models from scratch varies based on dataset size: a single GPU for the CIFAT-LT datasets, 2 GPUs for the ImageNet-LT dataset, and 4 GPUs for the iNaturalist dataset.

# D. More Experiment Results

## D.1. Additional Results on the CIFAR-LT Datasets

In this section, we show additional results on the CIFAR-LT datasets. Specifically, Tab.6 presents additional experimental results on the CIFAR-100 LT dataset with more combined methods. Tab.5 provides the experimental results on the CIFAR-10 LT dataset. The results suggest that Focal-SAM consistently outperforms SAM, ImbSAM, and CC-SAM across all methods and datasets, regardless of whether ResNet models are trained from scratch or foundation models are fine-tuned. This indicates that Focal-SAM offers better fine-grained control over the loss landscape for both head and tail classes, leading to improved overall performance. This further highlights the effectiveness of Focal-SAM.

Table 5: Performance comparison on CIFAR-10 LT datasets with various imbalance ratios (IR). FFT denotes fully fine-tuning the foundation model with LA loss.

| Method | IR100 | | | | IR200 | IR50 | IR10 |
|---|---|---|---|---|---|---|---|
| | Head | Med | Tail | All | All | All | All |
| Training from scratch | | | | | | | |
| CE | 87.0 | 73.6 | 54.0 | 73.1 | 68.6 | 78.3 | 87.4 |
| CE+SAM | **89.5** | 73.9 | 56.7 | 75.0 | 69.8 | 79.6 | 88.8 |
| CE+ImbSAM | 88.0 | **79.0** | 60.1 | 76.9 | **72.6** | 81.1 | 89.3 |
| CE+CC-SAM | 88.9 | 74.1 | 61.3 | 76.2 | 71.3 | 80.0 | 89.2 |
| **CE+Focal-SAM** | 89.3 | 75.4 | **62.9** | **77.2** | 71.7 | **82.0** | **90.0** |
| LDAM+DRW (Cao et al., 2019) | 85.5 | 74.6 | 69.0 | 77.3 | 73.8 | 80.8 | 87.3 |
| LDAM+DRW+SAM | **88.9** | 78.3 | 73.2 | 81.0 | 78.6 | **84.5** | 89.4 |
| LDAM+DRW+ImbSAM | 86.5 | **79.7** | 73.7 | 80.6 | 77.3 | 84.0 | 88.9 |
| LDAM+DRW+CC-SAM | 88.4 | 79.2 | 73.3 | 81.1 | 78.9 | 84.4 | 89.4 |
| **LDAM+DRW+Focal-SAM** | 88.7 | 79.5 | **74.2** | **81.6** | **79.2** | **84.5** | **89.5** |
| LA (Menon et al., 2021) | **87.6** | 72.6 | 70.1 | 77.9 | 74.3 | 81.6 | 87.8 |
| LA+SAM | 86.7 | 80.6 | 78.2 | 82.3 | 78.9 | 85.4 | 90.2 |
| LA+ImbSAM | 84.1 | **81.6** | 80.1 | 82.2 | 78.6 | 84.7 | 89.5 |
| LA+CC-SAM | 86.6 | 80.8 | 78.5 | 82.5 | 79.1 | **85.5** | 90.2 |
| **LA+Focal-SAM** | 86.9 | 81.2 | **79.2** | **82.9** | **79.6** | **85.5** | **90.5** |
| VS (Kini et al., 2021) | **88.1** | 77.1 | 68.4 | 78.9 | 74.7 | 81.5 | 88.3 |
| VS+SAM | 85.6 | **82.7** | 76.6 | 82.0 | 79.0 | 85.4 | 90.3 |
| VS+ImbSAM | 85.3 | 82.1 | 77.3 | 81.9 | 78.7 | 84.8 | 90.0 |
| VS+CC-SAM | 85.6 | 82.0 | 78.2 | 82.3 | 79.3 | 85.5 | 90.4 |
| **VS+Focal-SAM** | 87.7 | 80.6 | **78.8** | **82.9** | **79.5** | **85.8** | **90.7** |
| Fine-tuning foundation model | | | | | | | |
| FFT | **97.9** | 95.8 | 95.9 | 96.7 | 95.7 | 97.1 | 97.9 |
| FFT+SAM | 97.5 | 96.5 | 97.0 | 97.0 | **96.6** | 97.5 | 98.0 |
| FFT+ImbSAM | 97.0 | **97.0** | 97.4 | 97.1 | 96.5 | **97.7** | 97.9 |
| FFT+CC-SAM | 97.6 | 96.2 | 97.0 | 97.0 | **96.6** | 97.6 | 98.0 |
| **FFT+Focal-SAM** | 97.5 | 96.4 | **97.5** | **97.2** | **96.6** | 97.5 | **98.2** |
| LIFT (Shi et al., 2024) | **96.6** | 95.7 | 97.4 | 96.6 | 96.3 | 96.8 | 97.2 |
| LIFT+SAM | 96.6 | 95.6 | 97.8 | 96.7 | **96.4** | 96.7 | 97.0 |
| LIFT+ImbSAM | 96.5 | **95.9** | 97.7 | 96.7 | **96.4** | 96.7 | 97.2 |
| LIFT+CC-SAM | 96.5 | 95.6 | 97.9 | 96.6 | **96.4** | 96.7 | **97.3** |
| **LIFT+Focal-SAM** | **96.6** | 95.6 | **98.1** | **96.8** | **96.4** | **96.9** | **97.3** |

Table 6: Performance comparison on CIFAR-100 LT with more combined methods

| Method | IR100 | | | | IR200 | IR50 | IR10 |
| | Head | Med | Tail | All | All | All | All |
|---|---|---|---|---|---|---|---|
| Training from scratch | | | | | | | |
| LDAM+DRW (Cao et al., 2019) | 63.1 | 44.4 | 18.6 | 43.2 | 40.3 | 46.1 | 57.3 |
| LDAM+DRW+SAM | 67.6 | 51.7 | 25.9 | 49.5 | 45.8 | 52.6 | 61.1 |
| LDAM+DRW+ImbSAM | 62.5 | 48.8 | 26.4 | 46.9 | 42.5 | 51.3 | 59.8 |
| LDAM+DRW+CC-SAM | 66.5 | 52.2 | 26.2 | 49.4 | 45.7 | 52.3 | 61.0 |
| **LDAM+DRW+Focal-SAM** | **67.9** | **52.7** | **26.9** | **50.3** | **46.2** | **53.8** | **62.3** |
| VS (Kini et al., 2021) | 58.3 | 43.8 | **31.1** | 45.1 | 41.6 | 49.3 | 59.4 |
| VS+SAM | **62.7** | 52.0 | 29.3 | 49.0 | 45.5 | 53.5 | 62.5 |
| VS+ImbSAM | 56.1 | **53.3** | 29.9 | 47.2 | 44.7 | 52.6 | 62.6 |
| VS+CC-SAM | 62.2 | 52.2 | 30.3 | 49.1 | 45.2 | 53.7 | 62.9 |
| **VS+Focal-SAM** | **62.7** | 52.6 | 31.0 | **49.7** | **45.8** | **54.5** | **63.7** |

## D.2. CE and mCE Metrics on ImageNet-C for Long-tailed Domain Generalization

ImageNet-C (Hendrycks & Dietterich, 2019) contains the corrupted versions of ImageNet (Deng et al., 2009) dataset, with 15 corruption types applied at 5 severity levels, resulting in 75 dataset variations. In addition to the model's average accuracy across these 75 datasets, as shown in Tab.4, ImageNet-C introduces two additional metrics: Corruption Error (CE) and Mean Corruption Error (mCE). These metrics systematically assess the robustness of models against image corruption. CE measures the accuracy drop of a model on a specific type and severity of corruption compared to a baseline model, typically AlexNet (Krizhevsky et al., 2012). mCE aggregates the CE values across all corruption types and severity levels, providing a single robustness score for the model. Tab.7 presents the CE and mCE results on the ImageNet-C dataset when fine-tuning the foundation models. The results show that Focal-SAM generally achieves significantly lower CE and mCE values across the entire dataset and for each corruption type. This further demonstrates Focal-SAM's effectiveness in improving generalization.

Table 7: The CE and mCE values for different methods on the ImageNet-C dataset. The source models are trained on ImageNet-LT and evaluated on ImageNet-C. Lower values indicate better performance.

| Method | mCE↓ | Blur | | | | Noise | | | Digital | | | | Weather | | | |
| | | Motion | Defoc | Glass | Zoom | Gauss | Impul | Shot | Contr | Elast | JPEG | Pixel | Bright | Snow | Fog | Frost |
|---|---|---|---|---|---|---|---|---|---|---|---|---|---|---|---|---|
| FFT | 72.6 | 72.8 | 74.6 | 78.2 | 79.4 | 73.6 | 74.1 | 75.1 | 66.7 | 80.0 | 78.4 | 71.3 | 63.7 | 66.6 | 65.7 | 68.5 |
| +SAM | 69.0 | 69.2 | **72.2** | 76.9 | 76.9 | 68.9 | 69.7 | 70.7 | 63.6 | 77.5 | 74.5 | 66.8 | 59.2 | 62.6 | 60.6 | 65.7 |
| +ImbSAM | 69.5 | 69.4 | **72.2** | 76.8 | 77.1 | 70.2 | 70.7 | 72.0 | **62.9** | 77.6 | 76.4 | 68.5 | 59.6 | 62.5 | 60.2 | 65.7 |
| +CC-SAM | 69.0 | 69.0 | 74.1 | 76.6 | 77.8 | 69.5 | 69.3 | 71.2 | 64.0 | 76.6 | 75.2 | 67.0 | 58.1 | **61.3** | 59.6 | **65.0** |
| **+Focal-SAM** | **68.3** | **68.2** | 72.7 | **76.6** | **76.6** | **68.3** | **68.8** | **70.1** | 63.2 | **76.2** | 74.2 | **66.0** | 58.0 | 61.5 | 59.7 | 65.1 |
| LIFT | 63.6 | 61.7 | 67.6 | 75.9 | 72.4 | 61.0 | 61.2 | 62.7 | 54.1 | 80.8 | 72.7 | 60.3 | 52.1 | 57.0 | 52.2 | 62.1 |
| +SAM | 63.0 | 61.1 | 67.2 | 75.7 | 71.5 | 60.4 | 60.6 | 62.0 | 53.6 | 80.6 | 72.1 | 59.4 | 51.7 | 56.3 | 51.7 | 61.7 |
| +ImbSAM | 63.4 | 61.6 | 67.5 | 75.8 | 72.1 | 60.9 | 60.9 | 62.6 | 53.8 | 80.6 | 72.5 | 59.9 | 51.9 | 56.7 | 51.9 | 61.8 |
| +CC-SAM | 62.5 | 61.0 | 66.5 | 75.3 | **70.9** | 59.5 | **59.9** | 61.3 | 53.7 | 79.9 | 71.7 | **58.2** | 51.3 | 55.5 | 51.6 | 61.2 |
| **+Focal-SAM** | **62.2** | **60.6** | **66.3** | **75.2** | 71.1 | **59.3** | **59.9** | **61.0** | **53.1** | **79.6** | **71.0** | **58.2** | **50.7** | **55.3** | **51.1** | **61.0** |

## D.3. Results for Aligning Computational Cost

Focal-SAM requires about 50% more training time than SAM. To fairly evaluate the benefit of Focal-SAM, we conduct experiments where we extend the training epochs of SAM to match Focal-SAM's total computational cost. Specifically, we increase the training epochs to 300 or 30 (1.5 × the original 200 or 20) for SAM, while keeping Focal-SAM at 200 or 20 epochs. In this setting, the total computational cost of SAM and Focal-SAM becomes comparable. We conduct these experiments on CIFAR-100 LT, ImageNet-LT, and iNaturalist datasets. The results are shown in Tab.8, Tab.9, and Tab.10.

Table 8: Performance comparison on CIFAR-100 LT with aligned computational cost.

| Method | Epoch | IR100 All | IR200 All | IR50 All | IR10 All |
|---|---|---|---|---|---|
| Training from scratch | | | | | |
| CE+SAM | 300 | 43.0 | 39.2 | 46.9 | 60.0 |
| **CE+Focal-SAM** | 200 | **44.0** | **39.6** | **48.1** | **60.9** |
| LDAM+DRW+SAM | 300 | **50.4** | **46.4** | 53.0 | 61.2 |
| **LDAM+DRW+Focal-SAM** | 200 | 50.3 | 46.2 | **53.8** | **62.3** |
| VS+SAM | 300 | 49.2 | 45.5 | 53.0 | 63.3 |
| **VS+Focal-SAM** | 200 | **49.7** | **45.8** | **54.5** | **63.7** |
| LA+SAM | 300 | 50.1 | 45.5 | 53.8 | 63.0 |
| **LA+Focal-SAM** | 200 | **50.7** | **46.0** | **54.5** | **63.8** |
| Fine-tuning foundation model | | | | | |
| FFT+SAM | 30 | 81.2 | 78.3 | 83.6 | 86.9 |
| **FFT+Focal-SAM** | 20 | **81.6** | **79.0** | **83.9** | **87.3** |
| LIFT+SAM | 30 | 82.1 | **80.2** | 83.1 | 85.2 |
| **LIFT+Focal-SAM** | 20 | **82.4** | 80.0 | **83.2** | **85.4** |

Table 9: Performance comparison on ImageNet-LT with aligned computational cost.

| Method | Epoch | ImageNet-LT | | | |
|---|---|---|---|---|---|
| | | Head | Medium | Tail | All |
| Training from scratch | | | | | |
| LA+SAM | 300 | 63.2 | 51.6 | **34.8** | 53.8 |
| **LA+Focal-SAM** | 200 | **63.9** | **52.2** | 34.4 | **54.3** |
| Fine-tuning foundation model | | | | | |
| FFT+SAM | 30 | 80.6 | 73.1 | **56.1** | 73.6 |
| **FFT+Focal-SAM** | 20 | **80.8** | **73.9** | 54.4 | **73.9** |
| LIFT+SAM | 30 | **79.8** | 76.1 | 73.5 | 77.2 |
| **LIFT+Focal-SAM** | 20 | 79.7 | **76.6** | **73.6** | **77.4** |

Table 10: Performance comparison on iNaturalist with aligned computational cost.

| Method | Epoch | iNaturalist | | | |
|---|---|---|---|---|---|
| | | Head | Medium | Tail | All |
| Training from scratch | | | | | |
| LA+SAM | 300 | 68.0 | 71.4 | 72.4 | 71.5 |
| **LA+Focal-SAM** | 200 | **68.4** | **72.0** | **72.5** | **71.8** |

## D.4. Visualization of Loss Landscape

Fig.6 and Fig.7 visualize the loss landscape for head and tail classes of ResNet models trained with SAM, ImbSAM, CC-SAM, and Focal-SAM on the CIFAR-100 LT and CIFAR-10 LT datasets using VS loss respectively. From the results, we can observe that the loss landscape for tail classes with ImbSAM generally appears flatter and smoother than with SAM, suggesting that ImbSAM better flattens the loss landscape for tail classes. However, the head class loss landscape with

ImbSAM is generally sharper than with SAM, indicating that ImbSAM's exclusion of all head classes from the SAM term can sharpen the loss landscape for head classes, which might reduce their generalization performance. In contrast, CC-SAM and Focal-SAM provide fine-grained class-wise control, leading to a flatter loss landscape for both head and tail classes.

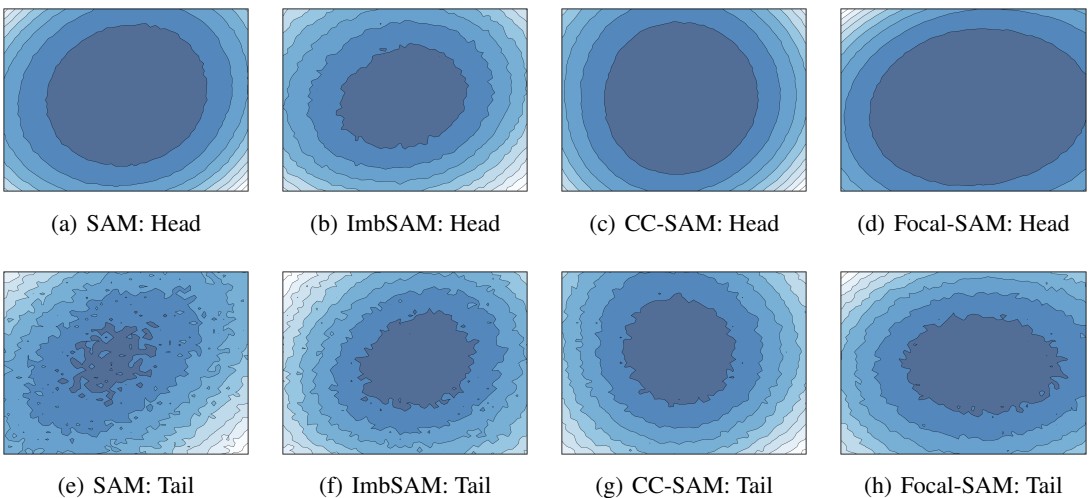

|  |  |  |  |
|---|---|---|---|
| (a) SAM: Head | (b) ImbSAM: Head | (c) CC-SAM: Head | (d) Focal-SAM: Head |
| (e) SAM: Tail | (f) ImbSAM: Tail | (g) CC-SAM: Tail | (h) Focal-SAM: Tail |

Figure 6: Visualization of loss landscape for head and tail classes of ResNet models trained with SAM, ImbSAM, CC-SAM, and Focal-SAM on CIFAR-100 LT using VS loss respectively.

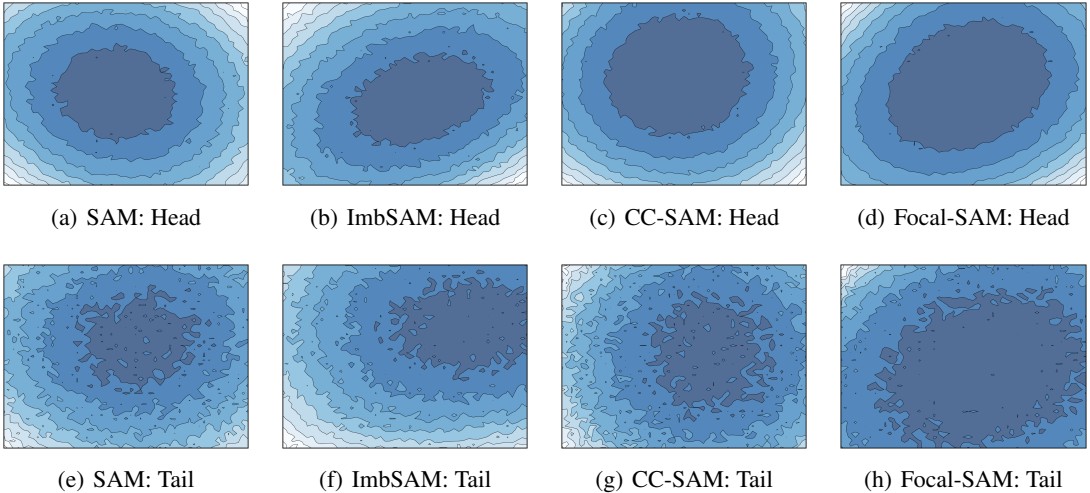

|  |  |  |  |
|---|---|---|---|
| (a) SAM: Head | (b) ImbSAM: Head | (c) CC-SAM: Head | (d) Focal-SAM: Head |
| (e) SAM: Tail | (f) ImbSAM: Tail | (g) CC-SAM: Tail | (h) Focal-SAM: Tail |

Figure 7: Visualization of loss landscape for head and tail classes of ResNet models trained with SAM, ImbSAM, CC-SAM, and Focal-SAM on CIFAR-10 LT using VS loss respectively.

## D.5. Ablation Study About Perturbation Radius $\rho$

Fig.8 illustrates the impact of the hyperparameter $\rho$ on the performance of Focal-SAM when combined with LDAM+DRW, LA, and VS methods on the CIFAR-LT datasets during ResNet models training. As $\rho$ increases, Focal-SAM's performance initially improves but then declines. This indicates a trade-off between achieving flatter minima and reducing training loss. The optimal value of $\rho$ for Focal-SAM is approximately $0.3$, which is higher than the commonly optimal value for SAM on balanced training datasets, as reported by Foret et al. (2021). This observation is consistent with Rangwani et al. (2022), who suggest that a larger $\rho$ can enhance performance in long-tailed learning.

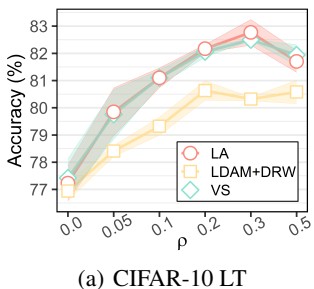 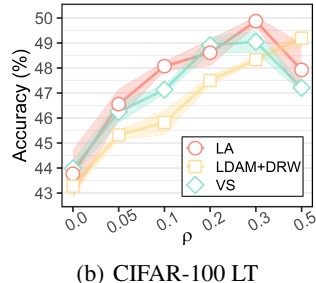

(a) CIFAR-10 LT

(b) CIFAR-100 LT

Figure 8: Ablation Study of Focal-SAM *w.r.t.* $\rho$

## D.6. Additional Results for Eigen Spectral Density of Hessian

This section presents additional results of the spectral density of hessian for ResNet models trained with SAM, ImbSAM, CC-SAM, and Focal-SAM. We analyze models trained on CIFAR-10 LT and CIFAR-100 LT datasets using VS and CE loss functions. The results are visualized in Fig.9, Fig.10 and Fig.11.

The results indicate that the largest eigenvalue $\lambda_{max}$ and the trace $tr(H)$ of the Hessian for tail classes are generally smaller with ImbSAM than with SAM. This suggests that ImbSAM flattens the loss landscape for tail classes more effectively. However, $\lambda_{max}$ and $tr(H)$ for head classes are typically larger with ImbSAM than with SAM, indicating that ImbSAM's coarse-grained strategy of excluding all head classes from SAM terms sharpens the loss landscape for those classes. In contrast, CC-SAM applies finer control over the loss landscape by using class-dependent perturbation radii, generally achieving lower $\lambda_{max}$ and $tr(H)$ for head and tail classes. Overall, both $\lambda_{max}$ and $tr(H)$ for head and tail classes are relatively lower with Focal-SAM than other SAM-based methods. This further suggests that Focal-SAM provides fine-grained control over the loss landscape, leading to a flatter landscape for both head and tail classes.

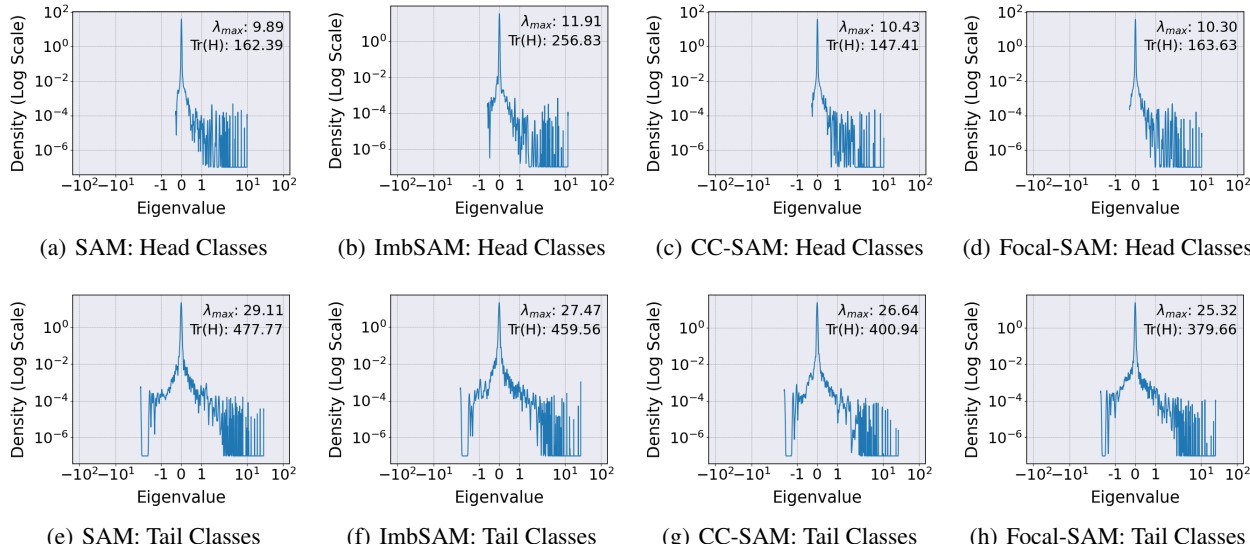

(a) SAM: Head Classes    (b) ImbSAM: Head Classes    (c) CC-SAM: Head Classes    (d) Focal-SAM: Head Classes

(e) SAM: Tail Classes    (f) ImbSAM: Tail Classes    (g) CC-SAM: Tail Classes    (h) Focal-SAM: Tail Classes

Figure 9: Eigen Spectral Density of Hessian for head and tail classes of ResNet models trained with SAM, ImbSAM, CC-SAM, and Focal-SAM on CIFAR-100 LT using VS loss respectively. A smaller $\lambda_{max}$ and $Tr(H)$ generally indicate a flatter loss landscape.

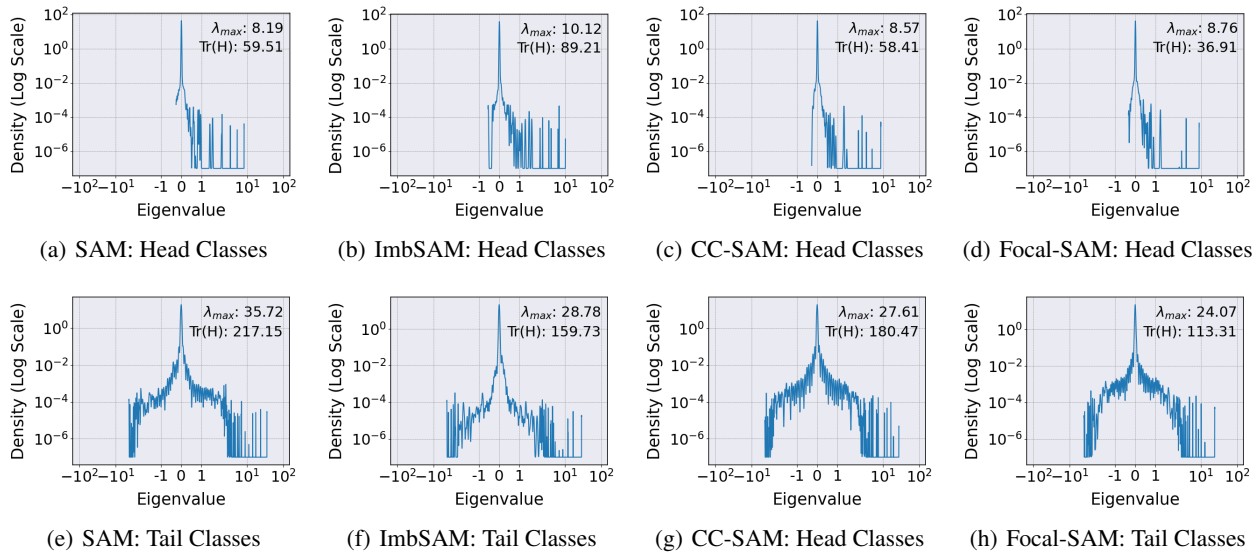

Figure 10: Eigen Spectral Density of Hessian for head and tail classes of ResNet models trained with SAM, ImbSAM, CC-SAM, and Focal-SAM on CIFAR-10 LT using CE loss respectively. A smaller $\lambda_{max}$ and $Tr(H)$ generally indicate a flatter loss landscape.

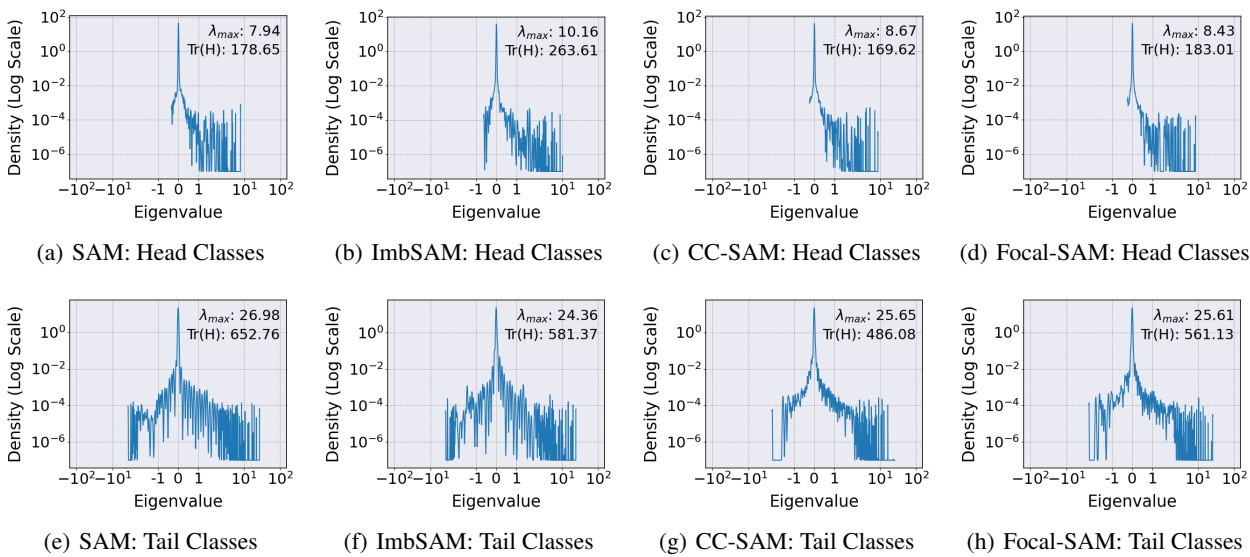

Figure 11: Eigen Spectral Density of Hessian for head and tail classes of ResNet models trained with SAM, ImbSAM, CC-SAM, and Focal-SAM on CIFAR-100 LT using CE loss respectively. A smaller $\lambda_{max}$ and $Tr(H)$ generally indicate a flatter loss landscape.

