# OpenReview forum: "Focal-SAM: Focal Sharpness-Aware Minimization for Long-Tailed Classification"
_ICML.cc/2025/Conference — ICML 2025 poster_

### Official Review · Reviewer_bvws · 2025-02-27

**Overall Recommendation:** 3

**Summary:**

The paper proposes a new sharpness aware minimization (SAM) algorithm suited for long-tailed classification. It works by formulating SAM as a regularizer, and then applying different regularization strengths for each class. The authors show that this approach can be more computationally efficient than baselines. The authors also show theoretical generalization bounds on their approach. Experiments on standard benchmarks empirically validate the effectiveness of the method.

**Claims And Evidence:**

Yes, all claims are sufficiently supported both theoretically and empirically.

**Essential References Not Discussed:**

No; as far as I know all relevant literature is discussed.

**Experimental Designs Or Analyses:**

The experimental protocol described in section 4.1 seems valid.

**Methods And Evaluation Criteria:**

Methods and evaluation criteria are well chosen; the authors evaluate on 4 different benchmarks including ImageNet-LT.

**Other Comments Or Suggestions:**

Please include standard errors over multiple runs for numerical results tables. The performance of Focal-SAM is often quite close to baselines, so establishing this statistical significance will be important.

**Other Strengths And Weaknesses:**

The paper is overall well-written and presented with solid empirical and theoretical results.

On the other hand, the idea proposed in this paper is a fairly straightforward combination of existing ideas (focal-loss and SAM). Moreover, this paper will likely be of interest mostly to the community working on long-tailed classification. Both of these limit the overall potential impact of the paper.

**Questions For Authors:**

1. How much do the numerical results vary between runs?

**Relation To Broader Scientific Literature:**

The two key related works are ImbSAM and CC-SAM. ImbSAM divides classes into head and tail groups, unlike Focal-SAM which does not divide the classes. CC-SAM applies SAM individually to each class, making it much more computationally intensive than Focal-SAM which applies a single perturbation to the parameters.

**Theoretical Claims:**

Theoretical claims (specifically Theorem 3.1) appear correct and rely on standard techniques.

---

> ### Author Rebuttal · Authors · 2025-03-31
>
> > Q1: Please include standard errors over multiple runs for numerical results tables. The performance of Focal-SAM is often quite close to baselines, so establishing this statistical significance will be important.
>
> **A1**: Thanks for your valuable suggestion!
>
> In the original paper, we report the results using the same fixed random seed for all methods to ensure a fair and consistent comparison. Following your advice, we have now conducted additional experiments using **three different random seeds** to further assess the effectiveness of our method. Specifically, we report the **average accuracy** along with the **standard error** across these runs. All other experimental settings remain the same as those described in the paper. These additional experiments are conducted on the CIFAR-100 LT datasets.
>
> Furthermore, we perform a **statistical significance** analysis using the **Mann-Whitney U test** to compare the performance of Focal-SAM against baseline methods:
>
> - Baselines marked with $^{**}$ indicate that Focal-SAM outperforms the corresponding baseline with a p-value $\le 0.05$.
> - Baselines marked with $^*$ indicate that Focal-SAM outperforms the corresponding baseline with a p-value $\le 0.10$.
>
> The results shows that the average performance of Focal-SAM consistently surpasses other SAM-based methods across multiple runs. This further supports the effectiveness of the proposed Focal-SAM approach.
>
> -----
>
> **CIFAR-100 LT**
>
> | Method             | IR100               | IR200               | IR50                | IR10                |
> | ------------------ | ------------------- | ------------------- | ------------------- | ------------------- |
> | CE                 | 41.2$\pm$0.3$^{**}$ | 37.7$\pm$0.3$^{**}$ | 45.3$\pm$0.4$^{**}$ | 57.9$\pm$0.2$^{**}$ |
> | CE+SAM             | 41.9$\pm$0.2$^{**}$ | 38.9$\pm$0.2$^{**}$ | 46.5$\pm$0.4$^{**}$ | 59.8$\pm$0.1$^{**}$ |
> | CE+ImbSAM          | 42.8$\pm$0.2$^{**}$ | 38.8$\pm$0.3$^{**}$ | 47.7$\pm$0.2        | 60.2$\pm$0.3$^{**}$ |
> | CE+CC-SAM          | 43.0$\pm$0.3$^{*}$  | 39.1$\pm$0.1$^{**}$ | 47.5$\pm$0.2$^{*}$  | 60.2$\pm$0.2$^{**}$ |
> | **CE+Focal-SAM**   | **43.7**$\pm$0.3    | **39.7**$\pm$0.2    | **48.0**$\pm$0.3    | **60.7**$\pm$0.1    |
> | LA                 | 44.8$\pm$0.2$^{**}$ | 42.0$\pm$0.3$^{**}$ | 50.2$\pm$0.2$^{**}$ | 59.3$\pm$0.3$^{**}$ |
> | LA+SAM             | 49.6$\pm$0.4$^{**}$ | 45.1$\pm$0.4$^{*}$  | 53.1$\pm$0.2$^{**}$ | 62.6$\pm$0.2$^{**}$ |
> | LA+ImbSAM          | 47.6$\pm$0.3$^{**}$ | 43.6$\pm$0.2$^{**}$ | 52.4$\pm$0.1$^{**}$ | 62.0$\pm$0.3$^{**}$ |
> | LA+CC-SAM          | 50.1$\pm$0.1$^{**}$ | 45.4$\pm$0.2        | 53.4$\pm$0.4$^{**}$ | 62.9$\pm$0.2$^{**}$ |
> | **LA+Focal-SAM**   | **50.6**$\pm$0.2    | **45.8**$\pm$0.3    | **54.3**$\pm$0.2    | **63.5**$\pm$0.2    |
> | FFT                | 78.7$\pm$0.2$^{**}$ | 76.1$\pm$0.2$^{**}$ | 81.3$\pm$0.1$^{**}$ | 85.5$\pm$0.1$^{**}$ |
> | FFT+SAM            | 81.0$\pm$0.1$^{**}$ | 77.6$\pm$0.1$^{**}$ | 83.4$\pm$0.2$^{**}$ | 86.7$\pm$0.1$^{**}$ |
> | FFT+ImbSAM         | 80.4$\pm$0.2$^{**}$ | 77.3$\pm$0.1$^{**}$ | 81.7$\pm$0.1$^{**}$ | 86.5$\pm$0.2$^{**}$ |
> | FFT+CC-SAM         | 81.1$\pm$0.1$^{**}$ | 78.2$\pm$0.1$^{**}$ | 83.5$\pm$0.1$^{**}$ | 87.0$\pm$0.1$^{*}$  |
> | **FFT+Focal-SAM**  | **81.6**$\pm$0.1    | **78.8**$\pm$0.2    | **83.9**$\pm$0.1    | **87.3**$\pm$0.2    |
> | LIFT               | 81.9$\pm$0.1$^{*}$  | 79.7$\pm$0.1$^{**}$ | 83.0$\pm$0.2$^{*}$  | 85.1$\pm$0.1$^{*}$  |
> | LIFT+SAM           | 82.0$\pm$0.2        | 79.7$\pm$0.1$^{**}$ | 83.1$\pm$0.1        | 85.2$\pm$0.2        |
> | LIFT+ImbSAM        | 81.9$\pm$0.1$^{*}$  | 79.8$\pm$0.2        | **83.2**$\pm$0.1    | 85.2$\pm$0.1        |
> | LIFT+CC-SAM        | 82.0$\pm$0.1        | 79.8$\pm$0.1$^{*}$  | 83.1$\pm$0.1        | 85.2$\pm$0.1        |
> | **LIFT+Focal-SAM** | **82.2**$\pm$0.1    | **80.1**$\pm$0.2    | **83.2**$\pm$0.1    | **85.4**$\pm$0.2    |

---

### Official Review · Reviewer_ag1Z · 2025-03-03

**Overall Recommendation:** 3

**Summary:**

The paper introduces Focal-SAM, a new variant of Sharpness-Aware Minimization (SAM) designed for long-tailed classification. It aims to improve generalization for both head and tail classes by integrating the focal mechanism with SAM. Compared with baselines like ImbSAM and CC-SAM, Focal-SAM efficiently achieves flatness in both head and tail classes. The authors present a theoretical generalization bound with improved convergence rates. Moreover, they validate Focal-SAM’s effectiveness through extensive experiments.

##Update

The authors' response addresses my concerns.

**Claims And Evidence:**

Yes.

**Essential References Not Discussed:**

No.

**Experimental Designs Or Analyses:**

Yes, I check Section 4 and Appendix D, E.

 - What is the computational cost of baselines such as LA, FFT, and LIFT? If their cost is approximately half that of SAM variants, a fairer comparison would be to evaluate them with twice the number of training steps.

**Methods And Evaluation Criteria:**

**Method:**
- Why don't use $\tilde{L}_S^{FS}(\mathbf(w))$ (eq 5) as the main objective? Specifically, why manage $\tilde{L}_S^{FS}(\mathbf(w))$ as a penalty of standard loss as in eq 6, which will add an additional hyperparameter $\lambda$?
- Why perturbation is computed only on $\tilde{L}_S^{FS}(\mathbf(w))$ rather than $L_S^{FS}(\mathbf(w))$ in eq 7?

**Evaluation:** Empirical evaluation makes sense to me.

**Other Comments Or Suggestions:**

NA

**Other Strengths And Weaknesses:**

NA

**Questions For Authors:**

(Summarize from above)


1. Why don't use $\tilde{L}_S^{FS}(\mathbf(w))$ (eq 5) as the main objective? Specifically, why manage $\tilde{L}_S^{FS}(\mathbf(w))$ as a penalty of standard loss as in eq 6, which will add an additional hyperparameter $\lambda$?
2. Why perturbation is computed only on $\tilde{L}_S^{FS}(\mathbf(w))$ rather than $L_S^{FS}(\mathbf(w))$ in eq 7?
3. What is the computational cost of baselines such as LA, FFT, and LIFT? If their cost is approximately half that of SAM variants, a fairer comparison would be to evaluate them with twice the number of training steps.

**Relation To Broader Scientific Literature:**

The paper proposes a SAM variant specifically for long-tailed classification tasks.

**Theoretical Claims:**

Yes, I checked Appendix B.

---

> ### Author Rebuttal · Authors · 2025-03-31
>
> > Q1: Why don't use $\tilde{L}_S^{FS} (w)$ (eq 5) as the main objective? Specifically, why manage $\tilde{L}_S^{FS} (w)$ as a penalty of standard loss as in eq 6, which will add an additional hyperparameter $\lambda$?
>
> Thanks for your questions!
>
> The motivation behind SAM-based methods is that the geometry of the loss landscape is closely related to a model's generalization ability. Specifically, **flatter minima** in the loss landscape generally leads to better generalization compared to sharper ones. Therefore, SAM-based methods aim to **simultaneously minimize both the loss value and the sharpness** to seek out flatter minima.
>
> In our Focal-SAM method, $\tilde{L}_S^{FS}(w)$ is a term which **reflects the sharpness of loss landscape** at $w$. If we only minimize $\tilde{L}_S^{FS}(w)$ , we can generally obtain a model parameter $w$ with a flat loss landscape. However, this **does not necessarily minimize the actual loss value** $L_S(w)$, which may remain large and degrade model performance.
>
> Therefore, we incorporate the sharpness term $\tilde{L}_S^{FS}(w)$ as a penalty term to the standard loss value $L_S(w)$, weighted by a hyperparameter $\lambda$. This enables **simultaneously minimization of both sharpness and the loss value**, where $\lambda$ controls the importance of the sharpness term.
>
> - If $\lambda$ is very large, the objective function primarily minimizes $\tilde{L}_S^{FS}(w)$, neglecting $L_S(w)$.
> - If $\lambda=0$, the method reduces to minimizing only the standard loss $L_S(w)$.
>
> In Fig.5 of our paper, we conduct an ablation study on $\lambda$ to explore these scenarios. The result show that **both excessively small and large values of $\lambda$ lead to suboptimal performance**, while a moderate $\gamma \approx 0.8$ achieves the best results. This highlights the importance of balancing loss minimization and sharpness control.
>
>
>
>
>
> > Q2: Why perturbation is computed only on $\tilde{L}_S^{FS} (w)$ rather than $L_S^{FS}(w)$ in eq 7?
>
> Thanks for your questions!
>
> To clarify, let's first recall the definition of the objective function $L_S^{FS} (w)$:
> $$
> L_S^{FS} (w) = L_S(w) + \lambda \cdot \tilde{L}_S^{FS} (w)
> $$
>
> where
> $$
> \tilde{L}_S^{FS} (w) = \max _{|| \epsilon ||_2 \le \rho} \sum _{i=1}^C (1 - \pi_i)^\gamma \tilde{L}_S^i(w, \epsilon)
> $$
>
> As shown, the perturbation $\epsilon$ appears **only** in the second term $\tilde{L}_S^{FS}(w)$ and is **independent** of the first term $L_S(w)$.
>
>
> Therefore, when optimizing $L_S^{FS}(w)$, the perturbation is **computed exclusively over the inner maximization problem in $\hat{\epsilon}(w)$**. Specifically, we solve for the optimal perturbation $\hat{\epsilon}(w)$ as:
> $$
> \hat{\epsilon}(w) = \arg\max _{|| \epsilon ||_2 \le \rho} \sum _{i=1}^C (1 - \pi_i)^\gamma \tilde{L}_S^i(w, \epsilon)
> $$
>
>
>
> > Q3: What is the computational cost of baselines such as LA, FFT, and LIFT? If their cost is approximately half that of SAM variants, a fairer comparison would be to evaluate them with twice the number of training steps.
>
> **A3**: Thank you for your valuable suggestion!
>
> Indeed, the computational cost of baselines such as LA, FFT and LIFT is approximately half that of SAM-based variants. To provide a fairer comparison, we conduct additional experiments in which we extend the training epochs of these baseline methods to roughly match the total computational cost of the SAM-based methods. Specifically, we double the training epochs to **400 or 40** (2$\times$ compared to the original 200 or 20) for these baselines, while keeping the SAM-based methods at **200 or 20** epochs. All other experimental settings remain consistent with those reported in the paper. These experiments are conducted on both CIFAR-100 LT and ImageNet-LT datasets. The results are summarized below.
>
> The results clearly show that, under comparable computational budgets, Focal-SAM consistently outperforms the baseline methods across different datasets. This further demonstrates that Focal-SAM effectively enhances model generalization in long-tailed learning scenarios.
>
>
>
> Below are the results:
>
> -----
>
> **CIFAR-100 LT**
>
> |Method|Epoch|IR100|IR200|IR50|IR10|
> |-|-|-|-|-|-|
> |CE|400|41.0|37.7|45.8|57.0|
> |**CE+Focal-SAM**|200|**44.0**|**39.6**|**48.1**|**60.9**|
> |LA|400|45.2|42.3|49.9|58.7|
> |**LA+Focal-SAM**|200|**50.7**|**46.0**|**54.5**|**63.8**|
> |FFT|40|76.4|74.2|79.8|85.1|
> |**FFT+Focal-SAM**|20|**81.6**|**79.0**|**83.9**|**87.3**|
> |LIFT|40|82.0|**80.3**|82.9|85.1|
> |**LIFT+Focal-SAM**|20|**82.4**|80.0|**83.2**|**85.4**|
>
>
> -----
>
>
> **ImageNet-LT**
>
> |Method|Epoch|Head|Medium|Tail|All|
> |-|-|-|-|-|-|
> |LA|400|61.6|45.7|31.1|49.8|
> |**LA+Focal-SAM**|200|**63.9**|**52.2**|**34.4**|**54.3**|
> |FFT|40|78.9|69.1|51.9|70.5|
> |**FFT+Focal-SAM**|20|**80.8**|**73.9**|**54.4**|**73.9**|
> |LIFT|40|79.4|75.9|72.1|76.7|
> |**LIFT+Focal-SAM**|20|**79.7**|**76.6**|**73.6**|**77.4**|

---

### Official Review · Reviewer_JBZt · 2025-03-13

**Overall Recommendation:** 4

**Summary:**

This paper proposes a learning mechanism named Focal Sharpness-Aware Minimization (Focal SAM), which is an exquisite extension of SAM theories over long-tailed classification tasks. Compared with existing methods, the proposed Focal SAM excels at keeping the flatness of landspaces of both head and tail classes. Extensive experiments show that Focal SAM outperforms existing methods on most datasets.

**Claims And Evidence:**

This paper proposes Focal SAM, which is motivated by several shortcomings of ImbSAM and CC-SAM. The authors give empirical analysis and visualizations of these.  This paper also gives the generalization bound of Focal SAM based on a theorem, and concrete proofs to this theorem are attached to the appendix. Conclusively, all claims are well validated.

**Essential References Not Discussed:**

I found no essential references missed in this paper.

**Experimental Designs Or Analyses:**

Yes. The experiment design is appropriate. The authors conduct experiments over four datasets. Experiments show that their Focal-SAM outperforms existing methods. The authors also make necessary explanations for this.

**Methods And Evaluation Criteria:**

Yes. The proposed Focal SAM is a successful application of SAM to long-tailed classification with the combination of class-wise scaling control. This paper may be of interest to the research community,  bringing new insights and broadening horizons.

**Other Comments Or Suggestions:**

It seems that Focal SAM may not be very efficient. Also, I'm not quite sure about some proof details concerning PAC-Bayesian generalization. Considering all of the above, I tend to give this paper a weak reject. If the author can answer my doubts well, I am willing to raise my score.

**Other Strengths And Weaknesses:**

- Strengths:
1. The paper is well-written and easy to follow.
2. The motivation is clear, where authors analyze the shortcomings
3. The idea is quite simple, easy to implement and also superior. The authors give concrete theoretical and experimental analysis of their method.
4. The experiments are extensive and well-designed.

- Weaknesses:
1. I notice that the running time of Focal-SAM is 50% more than the original SAM. The authors think that it's negligible given the strong performance of Focal SAM, while I can't agree.  From Table. 2, I observe a significant performance enhancement with the original SAM, but the improvements from SAM to Focal-SAM are not obvious. This dilates the necessity of employing Focal SAM.
2. I think that a quite amount of theoretical contributions are provided in the appendix. This makes the conclusion of your theorem a little abrupt. I suggest the authors move several lemmas to the main paper.
3. Meanings that Figure 1.b and Table. 1 try to convey seems repeated.

**Questions For Authors:**

Can you provide more explanations for the efficiency of Focal SAM? With a trade-off between performance and efficiency, I feel that Focal SAM has no explicit advantage compared with SAM.

**Relation To Broader Scientific Literature:**

I think this paper contributes much to the research community. Although the methodology is intuitive, the authors give detailed proofs and experiments to validate its superiority. This makes their methods solid and easy to implement, illuminating the research community.

**Theoretical Claims:**

This paper proposes a theorem to show the generalization bound of their Focal SAM. To the best of my knowledge, I found no technical flaws in the proof. I'm not quite sure about the part concerning PAC-Bayesian generalization.

---

> ### Author Rebuttal · Authors · 2025-03-31
>
> > Q1:  (1) From Table. 2, I observe a significant performance enhancement with the original SAM, but the improvements from SAM to Focal-SAM are not obvious. (2) I notice that the running time of Focal-SAM is 50% more than the original SAM. This dilates the necessity of employing Focal SAM.
>
> **A1**: Thanks for your question!
>
> **(1) Performance Gains:** It is true that the improvement from SAM to Focal-SAM is generally smaller than that from the baseline to SAM. This is expected, as SAM itself is already an effective method, and Focal-SAM is proposed as a **refinement of SAM**. Therefore, the additional improvement is naturally smaller but **still consistent and meaningful**. We also conduct multiple runs and statistical significance analysis to further confirm this. Please see `A1` for Reviewer `bvws` for details.
>
>
>
> **(2) Computational Cost:** While Focal-SAM increases training time by around 50% compared to SAM, this **does not mean it is unnecessary**. To fairly assess its benefit, we conduct experiments where we extend the training epochs of SAM to match Focal-SAM's total computational cost. Specifically, we increase the training epochs to **300 or 30** (1.5$\times$ the original 200 or 20) for SAM, while keeping Focal-SAM at **200 or 20** epochs. **In this setting, the total computational cost of SAM and Focal-SAM becomes comparable.** All other experimental settings remain the same as in the main paper. We conduct these experiments on CIFAR-100 LT, ImageNet-LT, and iNaturalist datasets. The results are summarized below, from which we can observe **consistent improvement**.
>
> -----
>
> **CIFAR-100 LT**
>
> |Method|Epoch|IR100|IR200|IR50|IR10|
> |-|-|-|-|-|-|
> |CE+SAM|300|43.0|39.2|46.9|60.0|
> |**CE+Focal-SAM**|200| **44.0**|**39.6**|**48.1**|**60.9**|
> |LDAM+DRW+SAM|300|**50.4**|**46.4**|53.0|61.2|
> |**LDAM+DRW+SAM**|200|50.3|46.2|**53.8**|**62.3**|
> |VS+SAM|300|49.2|45.5|53.0|63.3|
> |**VS+Focal-SAM**|200|**49.7**|**45.8**|**54.5**|**63.7**|
> |LA+SAM|300|50.1|45.5|53.8|63.0|
> |**LA+Focal-SAM**|200|**50.7**|**46.0**|**54.5**|**63.8**|
> |FFT+SAM|30|81.2|78.3|83.6|86.9|
> |**FFT+Focal-SAM**|20|**81.6**|**79.0**|**83.9**|**87.3**|
> |LIFT+SAM |30| 82.1|**80.2**|83.1|85.2|
> |**LIFT+Focal-SAM**|20|**82.4**|80.0|**83.2**|**85.4**|
>
> -----
>
> **ImageNet-LT**
>
> |Method|Epoch|Head|Medium|Tail|All|
> |-|-|-|-|-|-|
> |LA+SAM|300|63.2|51.6|**34.8**|53.8|
> |**LA+Focal-SAM**|200|**63.9**|**52.2**|34.4|**54.3**|
> |FFT+SAM|30|80.6|73.1|**56.1**|73.6|
> |**FFT+Focal-SAM**|20|**80.8**|**73.9**|54.4|**73.9**|
> |LIFT+SAM|30|**79.8**|76.1|73.5|77.2|
> |**LIFT+Focal-SAM**|20|79.7|**76.6**|**73.6**|**77.4**|
>
> -----
>
>
> **iNaturalist**
>
> |Method|Epoch|Head|Medium|Tail|All|
> |-|-|-|-|-|-|
> |LA+SAM|300|68.0|71.4|72.4|71.5|
> |**LA+Focal-SAM**|200|**68.4**|**72.0**|**72.5**|**71.8**|
>
>
>
>
>
>
>
>
>
>
> > Q2: I think that a quite amount of theoretical contributions are provided in the appendix. This makes the conclusion of your theorem a little abrupt. I suggest the authors move several lemmas to the main paper.
>
> **A2**: Thanks for your valuable suggestion!
>
> Due to space constraints in the anonymous submission version, we included the proof sketch and the lemmas supporting Theorem 3.1 in the appendix. We acknowledge that this may make the conclusion of Theorem 3.1 appear abrupt. **In the future version, we will revise the manuscript and relocate the proof sketch along with several key lemmas, such as Lemma B.2 and Lemma B.4, into the main text.**
>
>
>
>
>
> > Q3: Meanings that Figure 1.b and Table. 1 try to convey seems repeated.
>
> **A3**: Thanks for your question!
>
> We acknowledge that there is some overlap between Figure 1(b) and Table 1 regarding the part of computational cost. However, **the key messages we intend to convey through them are different**. In Figure 1(b), our primary goal is to directly compare Focal-SAM and CC-SAM to demonstrate the **effectiveness of Focal-SAM**, showing that it achieves better performance with higher efficiency. In contrast, Table 1 is used to **highlight the limitations of CC-SAM and to motivate the development** of our method. Specifically, we compare CC-SAM other SAM-based methods—including SAM, ImbSAM, and Focal-SAM—and show that CC-SAM incurs higher computational costs than the others. This observation motivates our proposal of Focal-SAM as an efficient alternative.

---

> > ### Comment · Reviewer_JBZt · 2025-04-03
> >
> > I have reviewed the response and appreciate the author's efforts. The response clarifies my concern about the significance of performance enhancement and the efficiency of the proposed Focal-SAM method. Additionally, I understand the theoretical contributions and the difference between Figure 1.b and Table 1. Overall, I recognize the novelty and effectiveness of this work. Hence, I will raise my rating to 4.

---

> > > ### Author Response · Authors · 2025-04-04
> > >
> > > Thank you very much for your response and for increasing the score!

---

### Official Review · Reviewer_n3JG · 2025-03-14

**Overall Recommendation:** 4

**Summary:**

For long-tailed (imbalanced) classificaton, the authors propose
Focal-SAM that aims at class-wise SAM so that flatter minima are
found.  They first show that imbSAM, which applies SAM only to tailed
classes, can increase sharpness in the head classes.  They then show
that CC-CAM could be computational expensive due to one
backpropagation for each class.

The proposed Focal-SAM has class-wise SAM, however, not class-wise
perturbation (epsilon) for SAM.  That is, they have one perturbation
for all classes, which requires only xone backpropagation for all
classes.  For each class, they first calculate L(w+epsilon) - L(w).
Then, for the focal-SAM loss, they weight each class loss with (the
weight component of) focal loss, which is then maximized over epsilon.
The overall loss is the oringial loss plus the focal-SAM loss.  They
discussed that only 3 backpropagations are needed.

For empirical evaluation, they use 4 datasets and compare with 3
existing SAM-based techniques, combine with and 3 methods for
imbalanced data.  Empirical results indicate that the proposed method
compares favorably.

## update after rebuttal

After reading and responding to the authors' rebuttal, I decided to maintain my rating.

**Claims And Evidence:**

The emiprical results indicate that Focal-SAM can achieve higher
accuracy with less computation.

**Essential References Not Discussed:**

I am not aware of essential references related to SAM within the
imbalanced context that are not discussed.

**Experimental Designs Or Analyses:**

Due to not being familiar with some of the terms, I did not check the
theoretical claim and proofs in the appendix.

**Methods And Evaluation Criteria:**

The proposed method is based on insights on the limitatons of imbSAM
CC-SAM.  By reducing sharpness for all classes and the number of
backpropagations, Focal-Sam can outperform imbSAM and CC-SAM.  Focal
loss is an existing method.  The evaluation criteria are reasonable.

**Other Comments Or Suggestions:**

see questions below

**Other Strengths And Weaknesses:**

The proposed method on class-wise SAM with fewer backpropagations is
interesting.  The empirical results indicate performance improvements
in accuracy`and speed.  The paper is generally well written.

**Questions For Authors:**

1.  On computational overhead, since a key difference is one
perturbation (epsilon) for all classes (Focal-SAM) vs for each class
(CC-SAM), how different are the perturbations between the two methods?

2.  Eq 4: L(w + epsilon) - L(w).  The second term does not seem to be
in imbSAM or CC-SAM.  Can you provide some discussion on the
difference?

3.  Figure 4: Gamma seems to be quick small (close to zero) for the
highest accuracy. That is, the focal component seems to be an
insignificant contributor. Any further insights on why?

4.  Figure 3: A lower $\pi_i$ has a higher $(1-\pi_i)^\gamma$, which implies a higher $(1-\pi_i)^\gamma$ (x-axis) has a lower probability density (y-axis).  However, this is not the case in the figure?

**Relation To Broader Scientific Literature:**

The key contributions are limited to SAM-based methods for long-tailed
classifications.

**Theoretical Claims:**

Due to not being familiar with some of the terms, I did not check the
theoretical claim and proofs in the appendix.

---

> ### Author Rebuttal · Authors · 2025-03-31
>
> > Q1:  On computational overhead, how different are the perturbations between the two methods?
>
> **A1:**  Thanks for your question!
>
> The perturbation in Focal-SAM is computed as:
> $$
> \hat{\epsilon}(w) = \rho \frac{\nabla_w L_S^\gamma(w)}{|| \nabla_w L_S^\gamma(w) ||_2}
> $$
>
> where $L_S^\gamma(w) = \sum_{i=1}^C (1 - \pi_i)^\gamma L_S^i(w)$.
>
> In contrast, CC-SAM computes a class-specific perturbation for each class $i$ as:
> $$
> \hat{\epsilon}_i(w) = \rho_i^* \frac{\nabla_w L_S^i(w)}{|| \nabla_w L_S^i(w) ||_2}
> $$
>
> **In terms of computational overhead:**
>
> Each perturbation requires one backpropagation (BP) to compute the corresponding gradient:
>
> - Focal-SAM requires only **one BP** to compute perturbation to compute $\nabla_w L_S^\gamma(w)$.
> - CC-SAM requires total **$C$ BPs**, one for each class, to compute $\nabla_w L_S^i(w)$.
>
> **The difference between the two perturbation are twofold:**
>
> - CC-SAM employs class-specific radii $\rho_i^*$ to finely control the sharpness penalty for each class. In contrast, Focal-SAM uses a unified perturbation radius $\rho$.
> - CC-SAM computes the gradient of class-wise loss $L_S^i(w)$ individually, while Focal-SAM computes the gradient of the weighted loss $L_S^\gamma(w)$.
>
> If we **disregard the difference in scaling across perturbations**, *i.e.*, assume:
> $$
> \frac{\rho}{|| \nabla_w L_S^\gamma(w) ||_2} = \frac{\rho_i^*}{|| \nabla_w L_S^i(w) ||_2}, \forall i
> $$
>
> then the Focal-SAM perturbation can be rewritten as:
> $$
> \hat{\epsilon}(w) = \sum_{i=1}^C (1 - \pi_i)^\gamma \hat{\epsilon}_i(w)
> $$
>
> This suggests that the perturbation in Focal-SAM can be viewed as a **weighted aggregation of the class-wise perturbations in CC-SAM**. Therefore, Focal-SAM implicitly considers the contribution of all classes and adjusts the emphasis across classes by tuning $\gamma$.
>
>
>
>
>
> > Q2: The second term in Eq.4 does not seem to be in ImbSAM or CC-SAM.
>
> **A2:** Thank you for your questions!
>
> In fact, we can **reformulate the objective functions of ImbSAM and CC-SAM to make the second term, $L(w)$, explicitly appear**. Furthermore, we can rewrite the objectives using the class-wise sharpness term (Eq.4).
>
> For ImbSAM:
> $$
> L_S^{IS}(w) = L_S^H(w) + L_S^T(w) + \max _{|| \epsilon ||_2 \le \rho} [L_S^T(w+\epsilon) - L_S^T(w)] = L_S(w) + \max _{|| \epsilon ||_2 \le \rho} \sum _{i \in T} \tilde{L}_S^i(w, \epsilon) \quad  (1)
> $$
>
> For CC-SAM:
> $$
> L_S^{CS}(w) = \sum_{i=1}^C \frac{1}{\pi_i} \cdot L_S^i(w) + \sum_{i=1}^C \max_{|| \epsilon || \le \rho_i^*} \frac{1}{\pi_i} \cdot [L_S^i(w + \epsilon) - L_S^i(w)] = \sum_{i=1}^C \frac{1}{\pi_i} \cdot L_S^i(w) + \sum_{i=1}^C \max_{|| \epsilon ||_2 \le \rho_i^*} \frac{1}{\pi_i} \cdot \tilde{L}_S^i(w, \epsilon) \quad (2)
> $$
>
> Eq.(1) shows that ImbSAM's objective includes the class-wise sharpness only for tail classes. This design specifically focuses on flattening the loss landscape for tail classes but may lead to poor generalization for head classes.
>
> Eq.(2) shows that CC-SAM considers the class-wise sharpness for all classes, using class-specific perturbation radii $\rho_i^*$. This allows it to more effectively flatten the loss landscape for both head and tail classes.
>
>
>
>
>
> > Q3: Fig.4: Gamma seems to be quite small (close to zero) for the highest accuracy.
>
> **A3:** Thanks for your questions! The reason is that **even a small $\gamma$ is sufficient to create a relatively skewed** $(1 - \pi_i)^\gamma$ from head to tail classes.
>
> As stated in the paper, we assume that $\pi_1 \ge \pi_2 \ge \cdots \ge \pi_C$. For example, in the CIFAR-10 LT dataset ($C = 10$ classes), when $\gamma = 0.8$ (the value at which the LA method achieves the best performance, as shown in Fig.4), we calculate that $(1 - \pi_{10})^\gamma$ is approximately $0.99$, whereas $(1 - \pi_1)^\gamma$ is around $0.66$. This indicates that even a relatively small $\gamma$ can result in a highly skewed $(1 - \pi_i)^\gamma$.
>
> Therefore, despite $\gamma$ being small, the focal component still plays a contributing role.
>
>
>
>
>
> > Q4: Figure 3: A lower $\pi$ has a higher $(1 - \pi_i)^\gamma$. However, this is not the case in the figure?
>
> **A4:** Thanks for your question! We apologize if our description of Fig.3 is unclear and potentially misleading.
>
> To clarify, Fig.3 presents the distribution of $(1 - \pi_i)^\gamma$ across various $\gamma$ and datasets. Specifically,  for a given $\gamma$ and dataset, we plot the distribution of  $\mathcal{P} = \\{ (1 - \pi_i)^\gamma \\} _ {i = 1} ^C$, where $C$ is the number of classes. Since the datasets **contain many tail classes with similarly small sample sizes** (*i.e.*, small $\pi_i$), the distribution **tends to peak at relatively large values of $(1 - \pi_i)^\gamma$**. Hence, Fig.3 differs from the scenario you expected, where it should plotting $\mathcal{P}' = \\{(1 - \pi_{y_n})^\gamma \\}_{n = 1}^N$, with $N$ being the number of samples.

---

### Decision · Program_Chairs · 2025-05-01

**Decision:**

Accept (poster)

**Comment:**

This paper proposes Focal-SAM, a class-wise variant of Sharpness-Aware Minimization tailored for long-tailed classification. The method combines the focal loss weighting scheme with a shared perturbation strategy to efficiently approximate class-wise SAM using only three backpropagations. The reviewers appreciated the clear motivation, simplicity of the approach, and its strong empirical performance across multiple datasets. While the improvement over standard SAM is modest and the theoretical contributions are partly relegated to the appendix, the method is well developed  and offers practical value for imbalanced classification settings. I recommend acceptance.